# Fast rates for prediction with limited expert advice

**El Mehdi Saad**[1], **Gilles Blanchard**[1,2]
[1]Laboratoire de Mathématiques d'Orsay, CNRS, Université Paris-Saclay; [2]Inria

## Abstract

We investigate the problem of minimizing the excess generalization error with respect to the best expert prediction in a finite family in the stochastic setting, under limited access to information. We assume that the learner only has access to a limited number of expert advices per training round, as well as for prediction. Assuming that the loss function is Lipschitz and strongly convex, we show that if we are allowed to see the advice of only one expert per round for $T$ rounds in the training phase, or to use the advice of only one expert for prediction in the test phase, the worst-case excess risk is $\Omega(1/\sqrt{T})$ with probability lower bounded by a constant. However, if we are allowed to see at least two actively chosen expert advices per training round and use at least two experts for prediction, the fast rate $\mathcal{O}(1/T)$ can be achieved. We design novel algorithms achieving this rate in this setting, and in the setting where the learner has a budget constraint on the total number of observed expert advices, and give precise instance-dependent bounds on the number of training rounds and queries needed to achieve a given generalization error precision.

**Keywords:** Online Learning, Budgeted Learning, Prediction with expert advice.

## 1 Introduction and setting

We consider a generic prediction problem in a stochastic setting: a target random variable $Y$ taking values in $\mathcal{Y}$ is to be predicted by a user-determined forecast $F$, also modeled as a random variable, taking values in a closed convex subset $\mathcal{X}$ of $\mathbb{R}^d$. The mismatch between the two is measured via a loss function $l(F, Y)$. The quality of the agent's output is measured by its generalization risk

$$R(F) := \mathbb{E}\big[l(F, Y)\big].$$

To assist us in this task, the forecast or "advice" of a number of "experts" $(F_1, \ldots, F_K)$ (also modeled as random variables) can be requested. The agent's objective is to achieve a risk as close as possible to the risk of the best expert $R^* = \min_{i \in [\![K]\!]} R(F_i)$ (for a nonnegative integer $n$, we denote $[\![n]\!] = \{1, \ldots, n\}$ ). We measure the performance of the user's forecast via its excess risk (or average regret) with respect to that best expert.

The literature on expert advice generally considers the *cumulative* regret over a sequence of forecasts $F_t$ followed by observation of the target variable $Y_t$ and incurring the loss $l(F_t, Y_t)$, $t = 1, \ldots, T$. In the present work we will separate observation (or training) phase and forecast phase: the user is allowed to observe (some of) the expert's predictions and the target variable for a number of independent, identically distributed rounds $(Y_t, F_{1,t}, \ldots, F_{K,t})_{1 \le t \le T}$ following certain rules to be specified. After the observation phase, the user must decide of a prediction strategy, namely a convex combination of the experts $\widehat{F} = \sum_{i=1}^k \widehat{w}_i F_i$, where the weights $\widehat{w}_i$ can be chosen based on the information gathered in the training phase. The risk of this strategy is $R(\widehat{F})$, where the risk is evaluated on new, independent data. In other words, if the training phase takes place over $T$ independent rounds, the forecast risk is the expected loss over the $(T + 1)$th, independent, round.

35th Conference on Neural Information Processing Systems (NeurIPS 2021).

In some situations, it may be overly expensive to query the advice of all experts at each round. The cost can be monetary if each expert demands to be paid to reveal his opinion, possibly because they have access to some information that others do not. In this case we may have a total limit on how much we can spend. In a different context, it is unrealistic to ask for the advice of all available doctors or to run a large battery of tests on each patient. In this case, we may be have a strong limit on the number of expert opinions that can be consulted for each training instance. In a more typical machine learning scenario, each "expert" might be a fixed prediction method $F_i = f_i(X)$ (using the information of a covariate $X$), where the predictor functions $f_i$ have been already trained in advance, albeit based on different sets of parameters or methodology; the goal then amounts to predictor selection or aggregation, in a situation where the computation of each single prediction constitutes the bottleneck cost, rather than data acquisition. Overall the agent's goal is to achieve a risk close to optimal while sparing on the number of experts queries – both at training time and for forecast.

Motivated by these questions we investigate several scenarios for prediction with limited access to expert advice. Furthermore, our emphasis is on obtaining *fast convergence rates* guarantees on the excess risk (i.e. $O(1/T)$ or $O(1/C)$, where $C$ is the total query budget). These are possible under a strong convexity assumption of the loss, specified below. Our contributions are the following.

- As a preliminary, we revisit (Section 3) the *full information setting*, with no limitations on queries. Maybe surprisingly, we contribute a new algorithm that is both simpler than existing ones and for which the proof of the fast convergence rate for excess risk is also elementary. Furthermore, for forecast we only need to consult 2 experts. The general principle of this algorithm will be reused in the limited observation settings.
- We then investigate (Section 4) the *budgeted setting* where we have a total query budget constraint $C$ for the training phase; then (Section 5) the *two-query* setting where the agent is limited to $m = 2$ queries per training round. In both cases, we give precise efficiency guarantees on the number of training expert queries needed to achieve a given precision for forecast. The obtained bounds come both in *instance-independent* (agnostic) and *instance-dependent* (depending on the experts' structure) flavors.
- Finally, we give some lower bounds (Section 6) were we show that fast rates cannot be achieved if the agent is only allowed to consult one single expert per training round *or* for forecast.

The following assumption on the loss will be made throughout the paper:

**Assumption 1.** $\forall y \in \mathcal{Y}$: $x \in \mathcal{X} \subseteq \mathbb{R}^d \mapsto l(x, y)$ is L-Lipschitz and $\rho$-strongly convex.

Recall that a function $f : \mathcal{X} \to \mathbb{R}$ is $L$-Lipschitz if $\forall x, y \in \mathcal{X}$:$|f(x) - f(y)| \leq L\|x - y\|$, and $\rho$-strongly convex if the function: $x \to f(x) - \frac{\rho^2}{2}\|x\|^2$ is convex.

**Remarks.** Assumption 1 implies that the diameter of $\mathcal{X}$ is bounded by $8L/\rho^2$ and the quantity $\sup_{x,x' \in \mathcal{X}, y \in \mathcal{Y}}|l(x, y) - l(x', y)|$ is bounded by $B := 8L^2/\rho^2$ (this notation shorthand will be used throughout the paper). Consequently, without loss of generality we can assume that the loss is bounded by $B$ (see Lemma[1] S-1 and subsequent discussion for details). It is satisfied, for example, in the following setting: least square loss $l(x, y) = (y - x)^2$ where $x \in \mathcal{X}$ and $y \in \mathcal{Y}$ with $\mathcal{X}$ and $\mathcal{Y}$ are bounded subsets of $\mathbb{R}^d$. Prior knowledge on $\rho$ is not necessary if $L$ and an upper bound on the the $l_\infty$ norm of the target variable $Y$ and the experts are known.

## 2 Discussion of related Work

**Games with limited feedback (slow rates):** Our work investigates what happens between the full information and single-point feedback games. Learning with a restricted access to information was considered under various settings in [6], [19], [12], [20], [5]. A setting close to ours was considered in [21], where the agent chooses in each round a subset of experts to observe their advice, then follows the prediction of one expert. To minimize the cumulative regret in the adversarial setting, they used an extension of the Exp3 algorithm, which allows to have an excess risk of $\mathcal{O}(\sqrt{1/T})$ in the limited feedback setting and $\mathcal{O}(\sqrt{\log(C)/C})$ in the budgeted case with a budget $C$.

---

[1]References starting with a prefix S- point to the supplemental material.

The differences in the setting considered here is that (a) we are interested in the generalization error in the stochastic setting rather than the cumulative regret in an adversarial setting and (b) our assumptions of the convexity of the loss allow for the possibility of fast excess risk convergence. Moreover, we consider the more general case where the player is allowed to combine $p$ out of $K$ experts for prediction. The possibility of playing a subset of arms was considered in the literature of Multiple Play Multi-armed bandits. It was treated with a budget constraint in [26] for example (see also [24]), where at each round, exactly $p$ out of $K$ possible arms have to be played. In addition to observing the individual rewards for each arm played, the player also learns a vector of costs which has to be covered with an a-priori defined budget $C$. In the stochastic setting, a UCB-type procedure gives a bound for the cumulative regret of $\mathcal{O}(\Delta_{\min}^{-1} \log(C)/C)$ that holds only in expectation, where $\Delta_{\min}^{-1}$ denotes the gap between the best choice of arms and the second best choice. This bound leads to an instance dependent bound of $\mathcal{O}(\sqrt{\log(C)/C})$ in the worst case. In the adversarial setting, an extension of Exp3 procedure gives a bound of $\mathcal{O}(\sqrt{\log(C)/C})$ for the cumulative regret that holds with high probability. In another online problem, where the objective is to minimize the cumulative regret in an adversarial setting with a small effective range of losses, [11] have shown the impossibility of regret scaling with the effective range of losses in the bandit setting, while [23] showed that it is possible to circumvent this impossibility result if the player is allowed one additional observation per round. However, in the settings considered, it is impossible to achieve a regret dependence on $T$ better than the rate of $\mathcal{O}(1/\sqrt{T})$.

**Fast rates in the full information setting:** The learning task of doing as well as the best expert of a finite family in the sense of generalization error has been studied quite extensively in the full information case. In an adversarial setting, it is well-known that under suitable assumptions on the loss function (typically related to strong convexity), an appropriately tuned exponential weighted average (EWA) strategy has cumulative regret bounded by the "fast rate" $\mathcal{O}(\log(K)/T)$ [13, 9, 4], which, combined with the online-to-batch conversion principle [8, 4] (also known as progressive mixture rule, [7, 25]), yields a bound of the same order for the *expected* excess prediction risk in the stochastic case. However, it was shown that progressive mixture type rules are *deviation suboptimal* for prediction [2], that is, their excess risk takes a value larger than $c/\sqrt{T}$ with constant positive probability over the training phase. To lift the apparent contradiction between the two last statements, consider that the excess risk of the EWA can take *negative* values, since it is an *improper* learning rule. Thus negative and positive "large" deviations can compensate each other so that the expectation is small. The inefficiency of EWA in deviation is a significant drawback, and alternatives to the EWA progressive mixture rule that achieve $\mathcal{O}(\log(K)/T)$ excess prediction risk with high probability were proposed by [17] and [3]. In [17], the strategy consists in whittling down the set of experts by elimination of obviously suboptimal experts, and performing empirical risk minimization (ERM) over the convex combinations of the remaining experts. In [3], the *empirical star* algorithm consists in performing an ERM over all segments consisting of a two-point convex combination of the ERM expert and any other expert. Note that the empirical star algorithm has the advantage that the final prediction rule is a convex combination of (at most) *two* experts.

**Linear regression with partially observed attributes:** Other related work is that of [10], and [14] on learning linear regression models with partially observed attributes. The most related setting to ours is the local budget setting, where the learner is allowed to output a linear combination of features for prediction. The key idea is to use the observed attributes in order to build an unbiased estimate of the full information sample, then to use an optimization procedure to minimize the penalized empirical loss. In our setting, the minimization of penalized empirical loss was shown to be suboptimal (see [16]). Moreover, while we want to predict as well as the best expert, in [10], the objective is to be as good as the best linear combination of features with a small additive term (the optimal rate, in this case, is $\mathcal{O}(1/\sqrt{T})$). Finally, we consider that the restriction on observed attributes (experts advice) does not apply only to the training samples but also to the testing data.

**Online convex optimization with limited feedback:** The idea of using multiple point feedback to achieve faster rates appeared in the online convex optimization literature (see [1], and [22]). It was shown that in the setting where the adversary chooses a loss function in each round if the player is allowed to query this function in two points, it is possible to achieve minimax rates that are close to those achievable in the full information setting. The key idea is to build a randomized estimate of the gradients, which are then fed into standard first-order algorithms. These ideas are not convertible into our setting because we consider a non-convex set of experts.

# 3 The full information case

In this section, we revisit the "classical" case where there is no constraint on the number of expert queries per observation round; assume the output of all experts are observed for $T$ rounds (in other words, $T$ i.i.d. training examples), which is the full information or "batch" setting. We want to output a final prediction rule with prediction risk controlled with high probability over the training phase.

We start with putting forward an apparently new rule , simpler than existing ones [17, 3], for the full information setting which, like the empirical star [3], outputs a convex combination of two experts. In contrast to the latter, our rule does not need any optimization over a union of segments. The underlying principle will guide us to construct a budget efficient expert selection rule in the sequel.

Define $\hat{R}(F_i) := T^{-1} \sum_{t=1}^{T} l(F_{i,t}, Y_t)$ the empirical loss of expert $i$, and $\hat{d}_{ij} := (T^{-1} \sum_{t=1}^{T} (F_{i,t} - F_{j,t})^2)^{\frac{1}{2}}$ the empirical $L_2$ distance between experts $i$ and $j$ over $T$ rounds. Finally let $\alpha = \alpha(\delta) := (\log(4K\delta^{-1})/T)^{\frac{1}{2}}$, where $\delta \in (0,1)$ is a fixed confidence parameter. Define

$$\Delta_{ij} := \hat{R}(F_j) - \hat{R}(F_i) - 6\alpha \max\left\{ L\hat{d}_{ij}, B\alpha \right\}. \tag{1}$$

The quantity $\Delta_{ij}$ can be interpreted as a test statistic: if $\Delta_{ij} > 0$, then we have a guarantee that $R(F_j) > R(F_i)$, so that expert $j$ is sub-optimal; this guarantee holds for all $(i,j)$ uniformly with probability $(1 - \delta)$. It therefore makes sense to reduce the set of candidates to

$$S := \left\{ j \in [\![K]\!] : \sup_{j \in [\![K]\!]} \Delta_{ij} \leq 0 \right\}. \tag{2}$$

Our new full information setting rule is the following:

$$\text{choose } \bar{k} \in S \text{ arbitrarily}; \quad \text{pick } \bar{j} \in \operatorname*{Arg\,Max}_{j \in S} \hat{d}_{\bar{k}j}; \quad \text{predict } \widehat{F} := \frac{1}{2}(F_{\bar{k}} + F_{\bar{j}}). \tag{3}$$

In words, the above rule consists in eliminating all experts that are manifestly outperformed by another one, and, among the remaining experts, pick two that disagree as much as possible (in terms of empirical $L^2$ distance ) and output their simple average for prediction. The next theorem establishes fast convergence rate for the excess risk of this rule:

**Theorem 3.1.** *If Assumption 1 holds and $\delta \in (0,1)$ is fixed, then for the prediction rule $\widehat{F}$ defined by* (3)*, it holds with probability $1 - 3\delta$ over the training phase ($c$ is an absolute constant):*

$$R(\widehat{F}) \leq R^* + cB\frac{\log(4K\delta^{-1})}{T}.$$

*Proof.* Let $d_{ij}^2 = \mathbb{E}\big[(F_i - F_j)^2\big]$. The result hinges on the following high confidence control of risk differences, established in Corollary S-4 as a direct consequence of the empirical Bernstein's inequality: with probability at least $1 - 3\delta$, it holds:

$$\text{For all } i, j \in [\![K]\!] : \qquad \Delta_{ij} \leq (R_j - R_i) \leq \Delta_{ij} + 32\alpha \max(Ld_{ij}, B\alpha). \tag{4}$$

Let $i^* \in \operatorname*{Arg\,Min}_{i \in [\![K]\!]} R_i$ be an optimal expert. Since $R_{i^*} - R_j \leq 0$ for all $j \in [\![K]\!]$, it follows that if (4) holds, then $i^* \in S$, from the definition of $S$. So if (4) holds, we have

$$R\left(\frac{F_{\bar{k}} + F_{\bar{j}}}{2}\right) \leq \frac{1}{2}\left(R_{\bar{k}} + R_{\bar{j}}\right) - \frac{\rho^2}{8} d_{\bar{k}\bar{j}}^2$$

$$= R^* + \frac{1}{2}\left((R_{\bar{k}} - R_{i^*}) + (R_{\bar{j}} - R_{i^*})\right) - \frac{\rho^2}{8} d_{\bar{k}\bar{j}}^2$$

$$\leq R^* + \frac{1}{2}\left(\Delta_{\bar{k}i^*} + \Delta_{\bar{j}i^*}\right) + 16\alpha\left(\max\left(Ld_{\bar{j}i^*}, B\alpha\right) + \max(Ld_{\bar{k}i^*}, B\alpha)\right) - \frac{\rho^2}{8} d_{\bar{k}\bar{j}}^2$$

$$\leq R^* + 32B\alpha^2 + 48L\alpha d_{\bar{k}\bar{j}} - \frac{\rho^2}{8} d_{\bar{k}\bar{j}}^2;$$

where we have used strong convexity of the loss (and therefore of $R(.)$ with respect to the $L^2$ distance) in the first line; the right-hand side of (4) in the third line; and, in the last line, the fact that $\bar{j}, \bar{k}, i^*$ are all in $S$ along with $d_{\bar{j}i^*} \leq d_{\bar{j}\bar{k}} + d_{\bar{k}i^*} \leq 2d_{\bar{j}\bar{k}}$ by construction of $\bar{j}$. Finally upper bounding the value of the last bound by its maximum possible value as a function of $d_{\bar{k}\bar{j}}$ and recalling $B = 8L^2/\rho^2$, we obtain the statement. $\qquad\square$

# 4 Budgeted Setting

In this section, we consider the budgeted setting. More precisely, given an a-priori defined budget $C$, at each round the decision-maker selects an arbitrary subset of experts and asks for their predictions. The choice of these experts may of course depend on past observations available to the agent. The player then pays a unit for each observed expert's advice. The game finishes when the budget is exhausted, at which point the player outputs a convex combination of experts for prediction.

We convert the batch rule defined in the full information setting to an "online" rule by performing the test $\Delta_{ji} > 0$ for each pair $(i, j)$ after each allocation. If at any round an expert $i \in [\![K]\!]$ fails any of these tests (i.e $\exists j : \Delta_{ji} > 0$), it is no longer queried. This extension allows us to derive instance dependent bounds, which cover the rates obtained in the batch setting in the worst case.

Since the tests $\Delta_{ij} > 0$ are performed after each allocation, we introduce the following modification on the definition of $\Delta_{ij}$, for concentration inequalities to hold uniformly over the runtime of the procedure. We define $\Delta_{ij}(t, \delta)$ as follows:

$$\Delta_{ij}(t, \delta) := \hat{R}(j, t) - \hat{R}(i, t) - 6\alpha(t, \delta/(t(t+1)) \max\left\{ L\hat{d}_{ij}(t), B\alpha(t, \delta/(t(t+1))) \right\}.$$

---

**Algorithm 1** Budgeted aggregation

---

**Input** $\delta$, $L$ and $\rho$.
Initialization: $S \leftarrow [\![K]\!]$.
**for** $T = 1, 2, \dots$ **do**
    Jointly query all the experts in $S$ and update $\Delta_{ij} > 0$ for all $i, j$.
    For all $i, j \in [\![K]\!]$, if $\Delta_{ij} > 0$, eliminate $j$: $S \leftarrow S \setminus \{j\}$.
    **if** the budget is consumed **then**
        let $\bar{k} \in S$, and $\bar{l} \leftarrow \underset{j \in S}{\arg\max}\ \hat{d}_{\bar{k}j}$.
        Return $\frac{1}{2}(F_{\bar{k}} + F_{\bar{l}})$.
    **end if**
**end for**

---

Let $\mathcal{S}^* := \text{Arg Min}_{i \in [\![K]\!]} R(F_i)$ denote the set of optimal experts. For $i, j \in [\![K]\!]$, we denote by $d_{ij} := (\mathbb{E}[(F_i - F_j)^2])^{1/2}$ the $L_2$ distance between the experts $F_i$ and $F_j$. For $i \in [\![K]\!]$, we introduce the following quantity:

$$\Lambda_i := \min_{i^* \in \mathcal{S}^*} \max\left\{ \frac{L^2 d_{ii^*}^2}{|R(F_i) - R(F_{i^*})|^2}; \frac{B}{R(F_i) - R(F_{i^*})} \right\}.$$

Define the following set of experts:

$$\mathcal{S}_\epsilon = \left\{ i \in [\![K]\!] : \Lambda_i > \frac{1}{\epsilon} \right\},$$

and let $\mathcal{S}_\epsilon^c$ be its complementary.

**Theorem 4.1.** *(Instance dependent bound) Suppose Assumption 1 holds. Let $C \geq K$ denote the global budget on queries and denote $\hat{g}$ the output of Algorithm 1 with inputs $(\delta, L, \rho)$ when the budget $C$ runs out. For any $\epsilon \geq 0$, if:*

$$C > 578 C_\epsilon \log\left( K\delta^{-1} C_\epsilon \right),$$

*where*

$$C_\epsilon := \sum_{i \in \mathcal{S}_\epsilon^c} \Lambda_i + |\mathcal{S}_\epsilon| \min\left\{ \frac{1}{\epsilon}; \Lambda^* \right\},$$

*where $\Lambda^* := \max_{i:\Lambda_i < +\infty} \Lambda_i$, then, with probability at least $1 - \delta$:*

$$R(\hat{g}) \leq R^* + cB\epsilon,$$

*where $c$ is an absolute constant.*

**Remarks.** *Observe that the above result gives in particular a query budget bound for the problem of best expert identification in our setting, by taking $\epsilon = 0$, in which case the required expert query budget is of order $\sum_{i:\Lambda_i < +\infty} \Lambda_i$ up to logarithmic terms. We can compare this to the problem of best arm identification in a bandit setting (one arm pull/query per round); our setting can be cast into that framework by considering each expert as an arm and only recording the information of the loss of the asked expert. The known optimal query bound for best arm identification in the classical multi-armed bandits setting with loss/reward bounded by $B$ is of order $\sum_{i:\Lambda_i < +\infty} \widetilde{\Lambda}_i$ [15], where $\widetilde{\Lambda}_i = B^2(R(F_i) - R(F_{i^*}))^{-2}$. Since the diameter of $\mathcal{X}$ is bounded by $B/L$ (see Lemma S-1), it holds $\Lambda_i \leq \widetilde{\Lambda}_i$. Hence, for best expert identification, the bound of Theorem 4.1 improves upon the best arm identification bound, potentially by a significant margin (in particular concerning the contribution of suboptimal but close to optimal experts for which $d_{ii^*} \ll B/L$ and $R_i - R_{i^*} \ll B$). Again, the improvement is due to the Assumption 1 on the loss and the possibility to query several experts per round, which are not used when casting the problem as a classical bandit setting.*

## 5  Two queries per round ($m = p = 2$)

In this section, we suppose that the decision-maker is constrained to see only two experts' advice per round ($m = 2$). We suppose that the horizon is unknown; when the game is halted, the player outputs a convex combination of at most two experts ($p = 2$). We will show that the rates obtained are as good as in the full information case in its dependence on the number of rounds $T$.

Algorithm 2 works as follows. To circumvent the limitation of observing only two experts per round, in each round, we sample a pair $(i, j) \in S \times S$ in a uniform way, where $S$ is the set of non-eliminated experts. Then the tests $\Delta'_{ji} \leq 0$ and $\Delta'_{ij} \leq 0$ are performed, where $\Delta'_{ij}$ is defined by (5). If $i$ or $j$ fail the test, which means that it is a suboptimal expert, it is eliminated from $S$.

Finally, when the algorithm is halted, depending on the number of allocated samples, we choose either an empirical risk minimizer over the non-eliminated experts or the mean of two experts from $S$ that are distant enough. This rule allows our algorithm's output to enjoy the best of converge rates of the two methods.

We introduce the following notations: In round $t$, denote $T_{ij}(t)$ the number of samples where predictions of experts $i$ and $j$ were jointly queried and $T_i(t)$ the number of rounds where the prediction of expert $i$ was queried. Denote $\hat{R}_{ij}(j, t)$ the empirical loss of expert $i$ calculated using only the $T_{ij}(t)$ samples queried for $(i, j)$ jointly. We define $\alpha_{ij}(t, \delta) := \sqrt{\frac{\log(4K\delta^{-1})}{T_{ij}(t)}}$ if $T_{ij}(t) > 0$ and $\alpha_{ij}(t) = \infty$ otherwise. Let $\hat{d}_{ij}(t)$ be the empirical $L_2$ distance between experts $i$ and $j$ based on the $T_{ij}(t)$ queried samples. Denote $\delta_t := \delta/(t(t+1))$. For $i, j \in [\![K]\!]$ we define:

$$\Delta'_{ij}(t, \delta) := \hat{R}_{ij}(j, t) - \hat{R}_{ij}(i, t) - 6 \max\Big\{ L\alpha_{ij}(t, \delta_t)\hat{d}_{ij}(t), B\alpha_{ij}^2(t, \delta_t) \Big\}. \tag{5}$$

---

**Algorithm 2** Two-point feedback

---

**Input** $\delta$, $L$ and $\rho$.
Initialization: $S \leftarrow [\![K]\!]$.
**for** $T = 1, 2, \ldots$ **do**
    Let $(i, j) \in \operatorname*{Arg\,Min}_{(u,v) \in S \times S} T_{uv}$.
    Query the advice of experts $i$ and $j$ and update the corresponding quantities.
    For all $u, v$: If $\Delta'_{uv} > 0$: $S \leftarrow S \setminus \{v\}$.
**end for**
**On interrupt:** Let $\hat{k} \in S$ and let $\hat{l} \leftarrow \operatorname*{argmax}_{j \in S} \hat{d}_{\hat{k}j}$.
Let $\hat{q}$ denote the empirical risk minimizer on $S$.
**if** $T_{\hat{k}\hat{l}} > \sqrt{\log(KT\delta^{-1})T_{\hat{q}}}$ **then**
    Return $\frac{1}{2}\big(F_{\hat{k}} + F_{\hat{l}}\big)$.
**else**
    Return $F_{\hat{q}}$.
**end if**

---

Our first result in this setting is an empirical bound. At any interruption time, it gives a bound on the excess risk, only depending on quantities available to the user, using the number of queries resulting from the querying strategy in Algorithm 2. We then use a worst-case bound on these quantities to develop an instance independent bound in Corollary 5.2.

**Theorem 5.1.** *(Empirical bound) Suppose Assumption 1 holds. Let $T \geq 2K^2$, and denote $\hat{g}$ the output of Algorithm 2 with inputs $(\delta, L, \rho)$ in round $T$. Then with probability at least $1 - 3\delta$:*

$$R(\hat{g}) \leq R^* + c\, B \min\left\{ \frac{\log(TK\delta^{-1})}{T_{\hat{k}\hat{l}}(T)}, \sqrt{\frac{\log(TK\delta^{-1})}{T_{\hat{q}}(T)}} \right\}, \qquad (6)$$

*where $\hat{k}, \hat{l}$ and $\hat{q}$ are the experts in Algorithm 2 and $c$ is an absolute constant.*

**Proof Sketch of Theorem 5.1** We start by noting that when running Algorithm 2, the optimal experts $\mathcal{S}^* = \operatorname{Arg\,Min}_{i \in [\![K]\!]} R(F_i)$ are never eliminated with high probability (Lemma S-5). This shows in particular, that when the procedure is terminated, we have $\mathcal{S}^* \subseteq S_T$, where $S_T$ is the set of non-eliminated experts at round $T$.

Then we show the following key result: in each round $t \leq T$, for any expert $i \in S_t$, let $j \in \operatorname{Arg\,Max}_{l \in S_t} \hat{d}_{il}(t)$, we have with probability at least $1 - \delta$:

$$R\left(\frac{F_i + F_j}{2}\right) \leq R^* + cB\, \frac{\log(K\delta_t^{-1})}{T_{ij}(t)}.$$

For the second bound, recall that $i^*$ belongs to $S_T$ with high probability. Therefore, performing an empirical risk minimization over the set of non-eliminated experts leads to the bound $\sqrt{\frac{\log(KT\delta^{-1})}{T_q(T)}}$, through a simple concentration argument using Hoeffding's inequality.

**Corollary 5.2.** *(Instance independent bound) Suppose assumption 1 holds. Let $T \geq 2K^2$, and denote $\hat{g}$ the output of Algorithm 2 with inputs $(\delta, L, \rho)$ in round $T$. Then with probability at least $1 - 3\delta$:*

$$R(\hat{g}) \leq R^* + c\, B \min\left\{ \frac{K^2 \log(TK\delta^{-1})}{T}, \sqrt{\frac{K \log(TK\delta^{-1})}{T}} \right\},$$

*where $c$ is an absolute constant.*

*Proof.* We develop an elementary bound on $T_{\hat{k}\hat{l}}$ and $T_{\hat{q}}$, then we inject these bounds into inequality (6).

Note that: $\hat{q}, i^* \in S_T$, hence $T_{\hat{q}}(T), T_{i^*}(T) \geq \frac{T}{2K}$. Moreover, we have:

$$T_{\hat{k}\hat{l}}(T) \geq \frac{T}{K^2}.$$

Using inequality (6), we obtain the result. $\qquad\square$

**Remarks.** *Observe that in all the considered settings (full information, budgeted and limited advice), the number of jointly sampled pairs $(F_i, F_j)$ to attain an excess risk of $\mathcal{O}(\epsilon)$ is of the order of $\mathcal{O}(K^2/\epsilon)$. Being able to ask a set of $m$ experts simultaneously in a training round allows to sample $m(m-1)/2$ pairs for a query cost of $m$: this is the advantage of the budgeted setting, while we have to query each pair in succession under the strict $m = 2$ constraint, resulting in a higher cost overall.*

**Theorem 5.3.** *(Instance dependent bound) Suppose Assumption 1 holds. Let $\hat{g}$ denote the output of Algorithm 2 with input $(\delta, L, \rho)$ and $T$ denote the total number of rounds. Let $\epsilon > 0$, if :*

$$T \geq 578\, C_\epsilon \log(\delta^{-1} C_\epsilon),$$

*where*

$$C_\epsilon := K \sum_{i \in \mathcal{S}_\epsilon^c} \Lambda_i + 2|\mathcal{S}_\epsilon|^2 \min\left\{\frac{1}{\epsilon}, \Lambda^*\right\},$$

*where $\Lambda^* := \max_{i : \Lambda_i < +\infty} \Lambda_i$, then, with probability at least $1 - \delta$:*

$$R(\hat{g}) \leq R^* + cB\, \epsilon,$$

*where $c$ is an absolute constant.*

**Remarks.** *If the algorithm is allowed to query $m > 2$ expert advices per round, then it can be modified to attain an improved excess risk. We present this extension in Section S-8 in the supplemental, and prove that it leads to a rate of $\mathcal{O}\left(\frac{(K/m)^2}{T}\log(KT/\delta)\right)$, which interpolates for intermediate values of $m$.*

**Proof Sketch of Theorem 5.3** First, we develop instance-dependent upper and lower bound for $T_{ij}(t)$, for any $i, j \in [\![K]\!]$ such that: $R(F_i) \neq R(F_j)$. To do this we introduce the following lemma (see Lemma S-7 in the supplemental):

**Lemma 5.4.** *Let $i, j \in [\![K]\!]$ such that $R(F_i) \neq R(F_j)$. With probability at least $1 - 4\delta$, for all $t \geq 1$, if*

$$T_{ij}(t) \geq 289 \log\big(K\delta_t^{-1}\big) \max\left\{ \frac{L^2 d_{ij}^2}{|R(F_i) - R(F_j)|^2} ; \frac{B}{|R(F_i) - R(F_j)|} \right\},$$

*then we have either $\Delta'_{ij} > 0$ or $\Delta'_{ji} > 0$; furthermore, if*

$$T_{ij}(t) \leq 3 \log\big(K\delta_t^{-1}\big) \max\left\{ \frac{L^2 d_{ij}^2}{|R(F_i) - R(F_j)|^2} ; \frac{B}{|R(F_i) - R(F_j)|} \right\},$$

*then we have: $\Delta'_{ij} \leq 0$ and $\Delta'_{ji} \leq 0$.*

This lemma gives in particular an upper bound on the number of allocations needed for an expert $i$ to be eliminated by an optimal expert $i^*$ (i.e. to fail the test $\Delta_{ii^*} \leq 0$). Then, we derive a bound on the number of rounds $T_\epsilon$ required to eliminate all the experts in $\bar{\mathcal{S}}_\epsilon^c$ and we conclude by showing that $T - T_\epsilon$ is large enough to ensure that the experts $\hat{k}$ and $\hat{l}$ in algorithm 2 satisfy $T_{\hat{k}\hat{l}} > 1/\epsilon$ with high probability.

# 6 Lower Bounds for $m = 1$ or $p = 1$

This section considers the case where the agent is restricted to selecting one expert at the end of the procedure ($p = 1$), and the case where the learner is restricted to see only one feedback per round ($m = 1$). We show that in either case it is impossible to do better than an excess risk $\mathcal{O}\big(1/\sqrt{T}\big)$ in deviation.

Lemma 6.1 is a direct consequence of a more general lower bound in [18], which proved that if the closure of the experts class is non-convex, and a single expert must be picked at the end ("proper" learning rule), then even under full information access during training the best achievable rate with high probability is $\mathcal{O}\big(1/\sqrt{T}\big)$.

**Lemma 6.1.** *($p = 1$) Consider the squared loss function. For $K = m = 2$ and $p = 1$, for any $T > 0$, and for any convex combination of the experts $\hat{g}$ output after $T$ training rounds, there exists a probability distribution for experts $\{F_1, F_2\}$ and target variable $Y$ (all bounded by 1) such that, with probability at least $0.1$,*

$$\hat{R}_T(\hat{g}) - R^* \geq \frac{c_1}{\sqrt{T}},$$

*where $c_1 > 0$ is an absolute constant.*

The second result shows that the same lower bound holds for the bandit feedback ($m = 1$) setting, even if the learner is allowed to predict using a convex combination of all the experts at the end. To the best of our knowledge, this is the first lower bound for deviations in this setting.

**Lemma 6.2.** *($m = 1$) Consider the squared loss function. For $K = p = 2$, and $m = 1$, for any $T > 0$, for any convex combination of the experts $\hat{g}$ output after $T$ training rounds, there exists a probability distribution for experts $\{F_1, F_2\}$ and target variable $Y$ (all bounded by 1) such that with probability at least $0.1$,*

$$\hat{R}_T(\hat{g}) - R^* \geq \frac{1}{2\sqrt{T}}.$$

# 7 Conclusion

We discussed the impact of restricted access to information in generalization error minimization with respect to the best expert. As many classical methods, such as progressive mixture rules (and randomized versions thereof) are deviation suboptimal, we proposed a new procedure achieving fast rates with high probability. We focused on the global budget setting, where a constraint on the total number of expert queries is made, and the local budget, where a limited number of expert advices are shown per round. Moreover, we proved fast rates are impossible to achieve if the agent is allowed to see just one expert advice per round or choose just one expert for prediction.

An interesting future direction is allowing experts to learn from data during the process. In this case, the i.i.d. assumption on the loss sequence is dropped, which necessitates deriving a new concentration for the key quantities.

**Acknowledgements**

We acknowledge support from the Agence Nationale de la Recherche (ANR), ANR-19-CHIA-0021-01 "BiSCottE"; and the Franco-German University (UFA) through the binational Doktorandenkolleg CDFA 01-18.

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
