# Supplementary Material for:
# Fast rates for prediction with limited expert advice

**El Mehdi Saad**[1]**, Gilles Blanchard**[1,2]
[1]Laboratoire de Mathématiques d'Orsay, CNRS, Université Paris-Saclay; [2]Inria

## 1 Notation

The following notation pertains to all the considered algorithms, where $t$ is a given training round:

- Let $\mathcal{T}_i(t)$ denote the set of training round indices where the advice of expert $i$ was queried and let $T_i(t) := |\mathcal{T}_i(t)|$.

- Let $\mathcal{T}_{ij}(t)$ denote the set of training round indices where the advice of experts $i$ and $j$ where jointly queried and let $T_{ij}(t) := |\mathcal{T}_{ij}(t)|$.

- Let $\hat{R}_{ij}(j,t)$ denote the empirical loss of expert $j$ calculated using only the $T_{ij}(t)$ samples queried for $(i,j)$ jointly:

$$\hat{R}_{ij}(j,t) := \frac{1}{T_{ij}(t)} \sum_{s \in \mathcal{T}_{ij}(t)} l(F_{j,s}, Y_s).$$

- $\hat{R}_i(t)$ denote the empirical loss of expert $i$ calculated using the $T_i(t)$ queried samples:

$$\hat{R}_i(t) := \frac{1}{T_i(t)} \sum_{s \in \mathcal{T}_i(t)} l(F_{i,s}, Y_s).$$

- Define $\alpha_{ij}(t,\delta) := \sqrt{\frac{\log(4K\delta^{-1})}{T_{ij}(t)}}$ if $T_{ij}(t) > 0$ and $\alpha_{ij}(t) = \infty$ otherwise.

- Define $\alpha_i(t,\delta) := \sqrt{\frac{\log(4K\delta^{-1})}{T_i(t)}}$ if $T_i(t) > 0$ and $\alpha_i(t) = \infty$ otherwise.

- Let $\hat{d}_{ij}(t)$ denote the empirical $L_2$ distance between experts $i$ and $j$ based on the $T_{ij}(t)$ queried samples:

$$\hat{d}_{ij}^2(t) := \frac{1}{T_{ij}(t)} \sum_{s \in \mathcal{T}_{ij}(t)} (F_{i,s} - F_{j,s})^2.$$

- Define $\Delta'_{ij}(t,\delta) := \hat{R}_{ij}(j,t) - \hat{R}_{ij}(i,t) - 6\alpha_{ij}(t,\delta) \max\left\{ L\hat{d}_{ij}(t), B\alpha_{ij}(t,\delta) \right\}$.

- Let $d_{ij}$ denote the $L_2$ distance between experts $i$ and $j$:

$$d_{ij} := \mathbb{E}\left[ (F_i - F_j)^2 \right].$$

- We denote $R(.)$ the expected risk function: $R(.) = \mathbb{E}[l(.,Y)]$, and define $R_i = R(F_i)$ for $i \in [\![K]\!]$.

35th Conference on Neural Information Processing Systems (NeurIPS 2021).

## 2   Some preliminary results

The lemma below shows that for a set $\mathcal{Y}$ and a convex set $\mathcal{X} \subseteq \mathbb{R}^d$, if there exists a function $l : \mathcal{X} \times \mathcal{Y} \to \mathbb{R}$ that is Lipschitz and strongly convex on its first argument, then the function $l$ and the set $\mathcal{X}$ are bounded.

**Lemma 1.** Let $\mathcal{X} \subseteq \mathbb{R}^d$ be a non-empty convex set, let $\mathcal{Y}$ be an arbitrary set and $l : \mathcal{X} \times \mathcal{Y} \to \mathbb{R}$ be a function such that for all $y \in \mathcal{Y} : l(., y)$ is $L$-Lipschitz and $\rho$-strongly convex, then we have:

- $\sup_{x,x' \in \mathcal{X}} \|x - x'\| \leq \frac{B}{L} = 8\frac{L}{\rho^2}$.

- $\sup_{x,x' \in \mathcal{X}, y \in \mathcal{Y}} |l(x, y) - l(x', y)| \leq B := 8\frac{L^2}{\rho^2}$

*Proof.* Let $y \in \mathcal{Y}$ and $x_0, x \in \mathcal{X}$, using the $\rho$-strong convexity of $l(., y)$ we have:

$$l\left(\frac{x + x_0}{2}, y\right) - \frac{\rho^2}{2}\left\|\frac{x + x_0}{2}\right\|^2 \leq \frac{1}{2}\left(l(x_0, y) - \frac{\rho^2}{2}\|x_0\|^2\right) + \frac{1}{2}\left(l(x, y) - \frac{\rho^2}{2}\|x\|^2\right)$$

Which implies:

$$\frac{\rho^2}{2}\left(\frac{1}{4}\|x_0 + x\|^2 - \frac{1}{2}\|x_0\|^2 - \frac{1}{2}\|x\|^2\right) \leq l\left(\frac{x + x_0}{2}, y\right) - \frac{l(x, y) + l(x_0, y)}{2}.$$

Using the parallelogram law and the assumption that $l$ is $L$-Lipschitz we have:

$$\frac{\rho^2}{8}\|x - x_0\|^2 \leq L\|x - x_0\|,$$

which proves that $\operatorname{diam}(\mathcal{X}) \leq 8\frac{L}{\rho^2}$. Now using the assumption that $l(., y)$ is $L$-Lipschitz, we have:

$$|l(x, y) - l(x_0, y)| \leq L\|x - x_0\|$$
$$\leq 8\frac{L^2}{\rho^2},$$

which proves the second claim. $\qquad\square$

For any $y \in \mathcal{Y}$, let $l^*(y) = \min_{x \in \mathcal{X}} l(x, y)$, which exists since $l$ is continuous in $x$ and $\mathcal{X}$ is a closed bounded set by the previous lemma, and let $\widetilde{l}(x, y) := l(x, y) - l^*(y)$. By the previous lemma, $\widetilde{l}(x, y) \in [0, B]$; also, note that the proposed algorithms remain unchanged if we replace the loss $l$ by $\widetilde{l}$, since the algorithms only depend on loss differences for different predictions $x, x'$ and the same $y$. Similarly, the excess loss of any predictor remains unchanged when replacing $l$ by $\widetilde{l}$. Therefore, without loss of generality we can assume that the loss function always takes values in $[0, B]$, which we do for the remainder of the paper.

The following lemma is technical, it will be used in the proof of the instance dependent bound (Theorem M-5.3).

**Lemma 2.** Let $x \geq 1, c \in (0, 1)$ and $y > 0$ such that:

$$\frac{\log(x/c)}{x} > y. \tag{1}$$

Then:

$$x < \frac{2\log\left(\frac{1}{cy}\right)}{y}.$$

*Proof.* Inequality (1) implies

$$x < \frac{\log(x/c)}{y},$$

and further

$$\log(x/c) < \log(1/yc) + \log\log(x/c) \leq \log(1/yc) + \frac{1}{2}\log(x/c),$$

since it can be easily checked that $\log(t) \leq t/2$ for all $t > 0$. Solving and plugging back into the previous display leads to the claim. $\qquad\square$

# 3 Some concentration results

In this section, we present concentration inequalities for the key quantities used in our analysis. Recall that Lemma 1 shows that under assumption M-1, without loss of generality we can assume that the loss function takes values in $[0, B]$, $B := 8L^2/\rho^2$.

The following lemma gives the main concentration inequalities we need:

**Lemma 3.** Suppose Assumption M-1 holds. For any integer $t \geq 1$, and $\delta \in [0, 1]$, with probability at least $1 - 3\delta$, for all $i, j \in [\![K]\!]$:

$$\left| \left( \hat{R}_{ij}(i, t) - \hat{R}_{ij}(j, t) \right) - (R_i - R_j) \right| \leq \sqrt{2} L \, \hat{d}_{ij} \, \alpha_{ij}(t, \delta) + 3B \, \alpha_{ij}^2(t, \delta)$$

$$\left| \hat{d}_{ij}^2 - d_{ij}^2 \right| \leq \max\left\{ 2\frac{B}{L} \, \alpha_{ij}(t, \delta) \, d_{ij} \; ; \; 6\left(\frac{B}{L}\right)^2 \alpha_{ij}^2(t, \delta) \right\}$$

$$\left| \hat{R}_i(t) - R_i \right| \leq 2B\alpha_i(t, \delta).$$

*Proof.* The first inequality is a direct consequence of the empirical Bernstein inequality (Theorem 4 in [3]). Recall that $l$ is $L$-Lipschitz in its first argument. Hence, we have the following bound on the empirical variance of the variable: $l(F_i, Y) - l(F_j, Y)$.

$$\widehat{\mathrm{Var}}[l(F_i, Y) - l(F_j, Y)] := \frac{2}{T_{ij}(t)(T_{ij}(t) - 1)} \sum_{u,v \in \mathcal{T}_{ij}(t)} \left( l(F_{i,u}, Y_u) - l(F_{j,u}, Y_u) - l(F_{i,v}, Y_v) + l(F_{j,v}, Y_v) \right)^2$$

$$\leq \frac{1}{T_{ij}(t)} \sum_{u \in \mathcal{T}_{ij}(t)} \left( l(F_{i,u}, Y_u) - l(F_{j,u}, Y_u) \right)^2$$

$$\leq L^2 \, \hat{d}_{ij}^2.$$

The second inequality is a consequence of Bernstein inequality applied to $\hat{d}_{ij}^2$, we used the following bound on the variance of the variable $(F_i - F_j)^2$:

$$\mathrm{Var}\left[ (F_i - F_j)^2 \right] \leq \mathbb{E}\left[ \|F_i - F_j\|^4 \right]$$

$$\leq \sup_{i,j \in [K]} \|F_i - F_j\|^2 \mathbb{E}\left[ \|F_i - F_j\|^2 \right]$$

$$\leq \left( \frac{B}{L} \right)^2 d_{ij}^2.$$

Finally, the last inequality stems from Hoeffding's inequality. $\qquad \square$

**Corollary 4.** Let $T > 0$ be fixed. In the full information case $(m = K)$, with probability at least $1 - 2\delta$, it holds:

$$\text{For all } i, j \in [\![K]\!] : \qquad \Delta_{ij} \leq (R_j - R_i) \leq \Delta_{ij} + 32\alpha \max(Ld_{ij}, B\alpha). \qquad (2)$$

*Proof.* In the full information case, since all experts are queried at each round we have $T_{ij}(T) = T_i(T) = T$ and $\alpha_{ij}(T, \delta) = \alpha(T, \delta) = \alpha$ for all $i, j$. Applying Lemma 3 in that setting, using the first inequality we obtain that with probability at least $1 - 3\delta$:

$$\Delta_{ij} \leq \left( \hat{R}(i, T) - \hat{R}(j, T) \right) - \sqrt{2} L\hat{d}_{ij}\alpha - 3B\alpha^2 \leq R_i - R_j,$$

giving the first inequality in (2); and

$$R_i - R_j \leq \left( \hat{R}(i, T) - \hat{R}(j, T) \right) + \sqrt{2} L\hat{d}_{ij}\alpha + 3B\alpha^2 \leq \Delta_{ij} + 9\alpha L\hat{d}_{ij} + 9B\alpha^2. \qquad (3)$$

From the second inequality in Lemma 3 we get, putting $\beta := B/L$:

$$\hat{d}_{ij}^2 - d_{ij}^2 \leq \max\{2\beta\alpha d_{ij}, 6\beta^2\alpha^2\}$$

$$\leq \max\left\{ 6\beta^2\alpha^2 + \frac{1}{6}d_{ij}^2, 6\beta^2\alpha^2 \right\}$$

$$\leq 6\beta^2\alpha^2 + \frac{1}{6}d_{ij}^2,$$

from which we deduce $\hat{d}_{ij}^2 \leq 12\alpha \max(\beta^2 \alpha^2, d_{ij}^2)$. Taking square roots and plugging into (3), we obtain the claim. $\qquad \square$

For $t \geq 1$, define: $\delta_t := \frac{\delta}{t(t+1)}$. Define the event $\mathcal{A}$:

$$(\mathcal{A}) : \forall t \geq 1, \forall\, i,j \in [\![K]\!] : \begin{cases} \left| \left( \hat{R}_{ij}(i,t) - \hat{R}_{ij}(j,t) \right) - (R_i - R_j) \right| \leq 3 \max \left\{ L\hat{d}_{ij}\, \alpha_{ij}(t, \delta_t); B\alpha_{ij}^2(t, \delta_t) \right\} & \text{(4a)} \\[2mm] \left| \hat{R}_i(t) - R_i \right| \leq 2B\, \alpha_i(t, \delta_t) & \text{(4b)} \\[2mm] \hat{d}_{ij}^2 \leq 12 \max \left\{ d_{ij}^2; \left( \frac{B}{L} \right)^2 \alpha_{ij}^2(t, \delta_t) \right\} & \text{(4c)} \\[2mm] d_{ij}^2 \leq 12 \max \left\{ \hat{d}_{ij}^2; \left( \frac{B}{L} \right)^2 \alpha_{ij}^2(t, \delta_t) \right\} & \text{(4d)} \end{cases}$$

Using a union bound over $t \geq 1$ and $i, j \in [\![K]\!]$, we have: $\mathbb{P}(\mathcal{A}) \geq 1 - 4\delta$.

## 4 Proof of Theorem M-5.1 and Corollary M-5.2

Let $t \geq 1$, denote by $S_t$ the set of non-eliminated experts in Algorithm M-2 at round $t$. The lemma below shows that conditionally to event $\mathcal{A}$, the best experts $\mathcal{S}^*$ are never eliminated.

**Lemma 5.** If $\mathcal{A}$ defined in (4) holds, $\forall t \geq 1$ we have: $\mathcal{S}^* \subseteq S_t$, where we recall $\mathcal{S}^* := \mathrm{Arg\,Min}_{i \in [\![K]\!]} R(F_i)$.

*Proof.* Let $t \geq 1$, assume for the sake of contradiction that: $i^* \in \mathcal{S}^*$ but $i^* \notin S_t$. Then, at some point, $i^*$ was eliminated by an expert $j$. More specifically: $\exists s \in [\![t]\!], \exists j \in [\![K]\!] \setminus \{i^*\}$, such that $\Delta'_{ji^*}(t, \delta_t) > 0$. It follows by definition of $\Delta'_{ji^*}$ that:

$$\hat{R}_{ji^*}(i^*, s) > \hat{R}_{ji^*}(j, s) + 6 \max \left\{ L\alpha_{ji^*}(s, \delta_s)\hat{d}_{ji^*}, B\alpha_{ji^*}^2(s, \delta_s) \right\}$$

which contradicts (4a) since we have: $R^* \leq R_j$. $\qquad \square$

The lemma below gives a high probability deviation rate on the excess of any expert in $S_t$ when combined with an appropriate expert. Recall that for $i \in [\![K]\!]$: $R_i = R(F_i)$.

**Lemma 6.** If event $\mathcal{A}$ defined in (4) holds, $\forall t \geq 1$, for all $i \in S_t$, let $j \in \mathrm{argmax}_{l \in S_t} \hat{d}_{il}(t)$, then we have:

$$R \left( \frac{F_i + F_j}{2} \right) \leq R^* + c\, B \frac{\log(K\delta_t^{-1})}{T_{ij}(t)},$$

where $c$ is an absolute constant.

*Proof.* Suppose that $\mathcal{A}$ is true. Let $t \geq 1$, $i \in S_t$ and $i^* \in \mathcal{S}^*$. Let $j \in \mathrm{argmax}_{S_t} \hat{d}_{il}$.

Lemma 5 shows that : $i^* \in S_t$, we therefore have by construction of Algorithm M-2:

$$\hat{R}_{ij}(j, t) \leq \hat{R}_{ij}(i, t) + 6 \max \left\{ L\alpha_{ij}(t, \delta_t)\hat{d}_{ij}(t), B\alpha_{ij}^2(t, \delta_t) \right\}$$

$$\hat{R}_{ii^*}(i, t) \leq \hat{R}_{ii^*}(i^*, t) + 6 \max \left\{ L\alpha_{ii^*}(t, \delta_t)\hat{d}_{ii^*}(t), B\alpha_{ii^*}^2(t, \delta_t) \right\}.$$

Using inequalities (4a) for $(i, j)$ and $(i, i^*)$ respectively and $\hat{d}_{ii^*}(t) \leq \hat{d}_{ij}(t)$, we have:

$$R_j \leq R_i + 9 \max \left\{ L\alpha_{ij}(t, \delta_t)\hat{d}_{ij}(t), B\alpha_{ij}^2(t, \delta_t) \right\} \tag{5}$$

$$R_i \leq R_{i^*} + 9 \max \left\{ L\alpha_{ii^*}(t, \delta_t)\hat{d}_{ij}(t), B\alpha_{ii^*}^2(t, \delta_t) \right\}. \tag{6}$$

We have:

$$R\left(\frac{F_i + F_j}{2}\right) \le \frac{1}{2}\left(R_i - \frac{\rho^2}{2}\mathbb{E}\big[F_i^2\big]\right) + \frac{1}{2}\left(R_j - \frac{\rho^2}{2}\mathbb{E}\big[F_j^2\big]\right) + \frac{\rho^2}{2}\mathbb{E}\left[\left(\frac{F_i + F_j}{2}\right)^2\right]$$

$$= \frac{1}{2}R_i + \frac{1}{2}R_j - \frac{\rho^2}{8}\left(2\mathbb{E}\big[F_i^2\big] + 2\mathbb{E}\big[F_j^2\big] - \mathbb{E}[(F_i + F_j)^2]\right)$$

$$= \frac{1}{2}R_i + \frac{1}{2}R_j - \frac{\rho^2}{8}d_{ij}^2$$

$$\le \frac{1}{2}R_i + \frac{1}{2}R_i + \frac{9}{2}\max\left\{L\alpha_{ij}(t,\delta_t)\hat{d}_{ij}(t), B\alpha_{ij}^2(t,\delta_t)\right\} - \frac{\rho^2}{8}d_{ij}^2$$

$$= R_i + \frac{9}{2}\max\left\{L\alpha_{ij}(t,\delta_t)\hat{d}_{ij}(t), B\alpha_{ij}^2(t,\delta_t)\right\} - \frac{\rho^2}{8}d_{ij}^2$$

$$\le R^* + \frac{27}{2}\max\left\{L\alpha_{ij}(t,\delta_t)\hat{d}_{ij}(t), B\alpha_{ij}^2(t,\delta_t)\right\} - \frac{\rho^2}{8}d_{ij}^2.$$

We used the strong convexity of $R$ in the first inequality and we injected (5) to bound $R(F_j)$ in the fourth line and (6) to bound $R(F_i)$ in the last line. Now we use inequality (4b) for $(i,j)$ and obtain:

$$R\left(\frac{F_i + F_j}{2}\right) - R^* \le 162\max\left\{L\alpha_{ij}(t,\delta_t)d_{ij}, B\alpha_{ij}^2(t,\delta_t)\right\} - \frac{\rho^2}{8}d_{ij}^2$$

$$\le c\,B\alpha_{ij}^2(t,\delta_t)$$

$$\le c\,B\alpha_{ij}^2(t,\delta_t),$$

where $c$ is an absolute constant. In the final step, we upper bounded the right-hand-side of the first inequality with a parabolic function in $d_{ij}$, then we replaced $d_{ij}$ with the expression achieving the maximum (recall that $B := 8(L/\rho)^2$).

$\square$

**Proof of Theorem M-5.1.** Let $T \ge 2K^2$, when Algorithm M-2 is halted at $T$. Let $\hat{k} \in S_T$ and $\hat{l} \in \text{argmax}_{j \in S_T}\hat{d}_{\hat{k}j}(T)$.

Let $\hat{q}$ denote the empirical risk minimizer on $S_T$:

$$\hat{q} \in \underset{j \in S_T}{\text{Arg Min}}\ \hat{R}_j(T).$$

We consider two cases. If $T_{\hat{k}\hat{l}}(T) > \sqrt{T_{\hat{q}}(T)\log(K\delta_T^{-1})}$, then the output of Algorithm M-2 is $\frac{F_{\hat{k}} + F_{\hat{l}}}{2}$ and we can apply the bound of Lemma 6.

If $T_{\hat{k}\hat{l}}(T) \le \sqrt{T_{\hat{q}}(T)\log(K\delta_T^{-1})}$, then the output of Algorithm M-2 is $F_{\hat{q}}$. We have:

$$R_{\hat{q}} - R_{i^*} = R_{\hat{q}} - \hat{R}_{\hat{q}}(T) + \hat{R}_{\hat{q}}(T) - \hat{R}_{i^*}(T) + \hat{R}_{i^*}(T) - R_{i^*}$$

$$\le 2B\sqrt{\frac{\log(K\delta_T^{-1})}{T_{\hat{q}}(T)}} + 2B\sqrt{\frac{\log(K\delta_T^{-1})}{T_{i^*}(T)}}$$

$$\le 2B\sqrt{\frac{\log(K\delta_T^{-1})}{T_{\hat{q}}(T)}} + 2B\sqrt{\frac{\log(K\delta_T^{-1})}{T_{\hat{q}}(T) - K}}$$

$$\le 5B\sqrt{\frac{\log(K\delta_T^{-1})}{T_{\hat{q}}(T)}},$$

where we used inequalities (4c) for $\hat{q}$ and $i^*$, and the fact that the allocation strategy leads to $|T_{i^*}(T) - T_{\hat{q}}(T)| \le K$ and $T_i(T) > 2K$ for all $i$.

As a conclusion we have:

$$R(\hat{g}) - R_{i^*} \le c\,B\min\left\{\frac{\log(KT\delta^{-1})}{T_{\hat{k}\hat{l}}(T)}; \sqrt{\frac{\log(KT\delta^{-1})}{T_{\hat{q}}(T)}}\right\}, \tag{7}$$

where $c$ is an absolute constant.

# 5  Proof of Theorem M-5.3

In this section, we prove instance dependent bounds on the number of rounds required to achieve a risk at least as good as the best expert up to $\epsilon > 0$.

The following lemma gives an instance dependent upper and lower bound on the quantities $T_{ij}(t)$, for $i, j \in [\![K]\!]$.

**Lemma 7.** Let $i, j \in [\![K]\!]$ such that $R_i \neq R_j$. If $\mathcal{A}$ holds, for all $t \geq 1$, if

$$T_{ij}(t) \geq 289 \log\big(K\delta_t^{-1}\big) \max\left\{ \frac{L^2 d_{ij}^2}{|R_i - R_j|^2} ; \frac{B}{|R_i - R_j|} \right\},$$

then we have either $\Delta'_{ij} > 0$ or $\Delta'_{ji} > 0$.

Furthermore, if

$$T_{ij}(t) \leq 3 \log\big(K\delta_t^{-1}\big) \max\left\{ \frac{L^2 d_{ij}^2}{|R_i - R_j|^2} ; \frac{B}{|R_i - R_j|} \right\},$$

then we have $\Delta'_{ij} \leq 0$ and $\Delta'_{ji} \leq 0$.

*Proof.* We start by proving the first claim of the lemma. Let $i, j \in [\![K]\!]$ and $t \geq 1$ such that:

$$T_{ij}(t) \geq 289 \log\big(K\delta_t^{-1}\big) \max\left\{ \frac{L^2 d_{ij}^2}{|R_i - R_j|^2} ; \frac{B}{|R_i - R_j|} \right\}. \tag{8}$$

Inequality (8) implies:

$$\alpha_{ij}(t, \delta_t) \leq \frac{1}{17} \min\left\{ \frac{|R_i - R_j|}{L d_{ij}} ; \sqrt{\frac{|R_i - R_j|}{B}} \right\}.$$

By simple calculus, we see that:

$$17 \, \max\big\{ L\alpha_{ij}(t, \delta_t) d_{ij} \; ; \; B\alpha_{ij}^2(t, \delta_t) \big\} \leq |R_i - R_j|.$$

Now we use inequality (4a) from event $\mathcal{A}$ to upper bound $|R_i - R_j|$:

$$17 \, \max\big\{ L\alpha_{ij}(t, \delta_t) d_{ij}; B\alpha_{ij}^2(t, \delta_t) \big\} \leq \Big| \hat{R}_{ij}(i, t) - \hat{R}_{ij}(j, t) \Big| + 3 \max\Big\{ L\alpha_{ij}(t, \delta_t)\hat{d}_{ij}(t); B\alpha_{ij}^2(t, \delta_t) \Big\}. \tag{9}$$

Using inequality (4b), we have:

$$\max\left\{ \hat{d}_{ij}(t); \frac{B}{L}\alpha_{ij}(t, \delta_t) \right\} \leq 2\sqrt{3} \max\left\{ d_{ij}; \frac{B}{L}\alpha_{ij}(t, \delta_t) \right\}.$$

We plug in the inequality above in (9) and obtain:

$$6 \max\Big\{ L\alpha_{ij}(t, \delta_t)\hat{d}_{ij}(t); B\alpha_{ij}^2(t, \delta_t) \Big\} < \Big| \hat{R}_{ij}(i, t) - \hat{R}_{ij}(j, t) \Big|,$$

implying that we have either $\Delta'_{ij}(t) > 0$ or $\Delta'_{ji}(t) > 0$.

For the second claim, Let $i, j \in [\![K]\!]$ and $t \in [\![T]\!]$ such that:

$$T_{ij}(t) \leq 3 \log\big(K\delta_t^{-1}\big) \max\left\{ \frac{L^2 d_{ij}^2}{|R_i - R_j|^2} ; \frac{B}{|R_i - R_j|} \right\}. \tag{10}$$

If $T_{ij}(t) = 0$, then $\Delta'_{ij} = \Delta'_{ji} = -\infty$.

Otherwise, inequality (10) implies that:

$$|R_i - R_j| \leq 3 \max\big\{ L\alpha_{ij}(t, \delta_t) d_{ij} \; ; \; B\alpha_{ij}^2(t, \delta_t) \big\}.$$

Now we use inequality (4a) from event $\mathcal{A}$ to lower bound $|R_i - R_j|$. We have:

$$\left|\hat{R}_{ij}(i,t) - \hat{R}_{ij}(j,t)\right| - 3\max\left\{L\alpha_{ij}(t,\delta_t)\hat{d}_{ij}(t) \; ; \; B\alpha_{ij}^2(t,\delta_t)\right\} \leq 3\max\left\{L\alpha_{ij}(t,\delta_t)d_{ij} \; ; \; B\alpha_{ij}^2(t,\delta_t)\right\}.$$

We plug in inequality (4d) to upper bound $d_{ij}$. We conclude that:

$$\left|\hat{R}_{ij}(i,t) - \hat{R}_{ij}(j,t)\right| \leq 6\max\left\{L\alpha_{ij}(t,\delta_t)\hat{d}_{ij}(t); B\alpha_{ij}^2(t,\delta_t)\right\},$$

implying that we have: $\Delta'_{ij}(t) \leq 0$ and $\Delta'_{ji}(t) \leq 0$. $\qquad\square$

Now we turn to the proof of Theorem M-5.3. Recall the following notations: for $i \in [\![K]\!]$ define:

$$\Lambda_i := \min_{i^* \in \mathcal{S}^*} \max\left\{\frac{L^2 d_{ii^*}^2}{|R_i - R_{i^*}|^2}; \frac{B}{R_i - R_{i^*}}\right\}.$$

Denote the corresponding reordered values:

$$\Lambda_{(1)} \leq \Lambda_{(2)} \leq \cdots \leq \Lambda_{(K)} = +\infty,$$

and $\Lambda^* := \min\{\Lambda_i; \Lambda_i < +\infty\}$.

**Proof of Theorem M-5.3.** By Lemma 6, in order to show that $R(\hat{g}) \leq R^* + cB\epsilon$, it suffices to prove that for any $i, j \in S_T$, it holds $T_{ij}(T) \geq B\log(K\delta_T^{-1})/\epsilon$.

Let $\epsilon > 0$, define the following sequences, for $N \in [\![K-1]\!]$:

$$\begin{cases} \phi_N & := 289(K-N)^2\big(\Lambda_{(N)} - \Lambda_{(N-1)}\big)\log\big(\delta^{-1}C_\epsilon\big); \\ \tau_N & := \sum_{k=1}^N \phi_k, \end{cases}$$

where we define $\Lambda_{(0)} = 0$ and

$$C_\epsilon := K\sum_{i \in \mathcal{S}_\epsilon^c} \Lambda_i + 2|\mathcal{S}_\epsilon|^2 \min\left\{\frac{1}{\epsilon}, \Lambda^*\right\}.$$

**Claim 1.** *If event $\mathcal{A}$ holds, for any $N \in [\![K]\!]$ after round $\lceil\tau_N\rceil$, all experts $i$ satisfying $\Lambda_i \leq \Lambda_{(N)}$ are necessarily eliminated.*

*Proof.* Recall that the number of queries required to eliminate an expert $i \in [\![K]\!]$ is upper bounded by the number of data points needed to have: $\Delta_{i^*i} > 0$ for any $i^* \in \mathcal{S}^*$, which would lead to the elimination of $i$ by $i^*$.

Let $i^*$ be an arbitrary element of $\mathcal{S}^*$. We use an induction argument, for $N = 1$ the claim is a direct consequence of the definition of $\tau_1$ and Lemma 7. Let $N < K$ and suppose that the claim is valid for all $i \leq N$. Let $j$ denote an expert such that $\Lambda_j = \Lambda_{(N+1)}$ and $j$ was not eliminated before $\lceil\tau_N\rceil$. For $i \leq N$, the induction hypothesis suggests that between round $\lceil\tau_i\rceil$ and $\lceil\tau_{i+1}\rceil$ there was at most $K - i$ non-eliminated experts. Since the allocation strategy is uniform over the pairs of experts in $S \times S$, we have:

$$T_{ji^*}(\tau_{N+1}) \geq 2\sum_{i=0}^N \frac{\tau_{i+1} - \tau_i}{(K-i)(K-i+1)}, \tag{11}$$

where $\tau_0 = 0$. Recall that the definition of $\tau_i$ implies that:

$$\tau_{i+1} - \tau_i = 289(K-i-1)^2\log\big(C_\epsilon\delta^{-1}\big)\big(\Lambda_{(i+1)} - \Lambda_{(i)}\big). \tag{12}$$

We plug in the lower bound given in (12) into (11) to obtain:

$$T_{ji^*}(\tau_{N+1}) \geq 289\log\big(C_\epsilon\delta^{-1}\big)\Lambda_{(N+1)}.$$

Using Lemma 7 we conclude that expert $j$ is eliminated before round $\tau_{N+1}$, which completes the induction argument.

$\qquad\square$

**Claim 2.** *We have for any $N \in [\![K]\!]$:*

$$\tau_N = 289 \log(C_\epsilon \delta^{-1}) \left( \sum_{i=1}^{N-1} (2(K-i)+1)\Lambda_{(i)} + (K-N)^2 \Lambda_{(N)} \right).$$

*Proof.* We have by definition of $\tau_N$:

$$\tau_N = \sum_{i=1}^{N} \phi_i$$

$$= \sum_{i=1}^{N} 289(K-i)^2 \left( \Lambda_{(i)} - \Lambda_{(i-1)} \right) \log(\delta^{-1} C_\epsilon)$$

$$= \sum_{i=1}^{N} 289(K-i)^2 \Lambda_{(i)} \log(\delta^{-1} C_\epsilon) - \sum_{i=1}^{N} 289(K-i)^2 \Lambda_{(i-1)} \log(\delta^{-1} C_\epsilon)$$

$$= 289 \log(\delta^{-1} C_\epsilon) \left( \sum_{i=1}^{N-1} (2(K-i)+1)\Lambda_{(i)} + (K-N)^2 \Lambda_{(N)} \right).$$

$\square$

**Conclusion:** Let $N_\epsilon$ denote the integer satisfying (we do not consider the trivial case where all the expert have the same risk):

$$\Lambda_{(N_\epsilon)} < \frac{1}{\epsilon} < \Lambda_{(N_\epsilon+1)}.$$

Recall that we suppose that $T$ satisfies:

$$T \geq 578 C_\epsilon \log(C_\epsilon \delta^{-1}).$$

Observe that (using Claim 2):

$$T \geq \tau_{N_\epsilon} + 289 \log(C_\epsilon \delta^{-1}) \left( 2|\mathcal{S}_\epsilon|^2 \min\left\{\frac{1}{\epsilon}; \Lambda^*\right\} - (K - N_\epsilon)^2 \Lambda_{(N_\epsilon)} \right) \qquad (13)$$

$$\geq \tau_{N_\epsilon} + 289 \log(C_\epsilon \delta^{-1}) \left( 2|\mathcal{S}_\epsilon|^2 \min\left\{\frac{1}{\epsilon}; \Lambda^*\right\} - |\mathcal{S}_\epsilon|^2 \Lambda^* \right) \qquad (14)$$

$$\geq \tau_{N_\epsilon} + 289 \log(C_\epsilon \delta^{-1}) |\mathcal{S}_\epsilon|^2 \min\left\{\frac{1}{\epsilon}; \Lambda^*\right\}. \qquad (15)$$

Claims 1 and 2 show that after $\lceil \tau_{N_\epsilon} \rceil$ rounds only elements $i \in [\![K]\!]$ satisfying: $\Lambda_i \leq \Lambda_{(N_\epsilon)}$ are eliminated. Therefore, if $1/\epsilon > \Lambda^*$, we have : $\Lambda_{(N_\epsilon)} = \Lambda^*$ and all the remaining experts are optimal (i.e. in $\mathcal{S}^*$). Hence the mean of any two experts in $\mathcal{S}$ satisfies: $R(\hat{g}) \leq R^*$.

Now suppose that $1/\epsilon < \Lambda^*$. We have for the last $T - \lceil \tau_{N_\epsilon} \rceil$ rounds all the experts in $\mathcal{S}_\epsilon^c$ were eliminated (hence there was at most $|\mathcal{S}_\epsilon|$ non-eliminated experts). Let $(\hat{k}, \hat{l})$ denote the pair output by algorithm M-2 after $T$ rounds, we have:

$$T_{\hat{k}\hat{l}}(T) \geq \log(C_\epsilon \delta^{-1}) \frac{T - \tau_{N_\epsilon}}{|\mathcal{S}_\epsilon|^2}$$

$$\geq 289 \frac{\log(C_\epsilon \delta^{-1})}{\epsilon}$$

$$\geq c \log(KT\delta^{-1}) \frac{1}{\epsilon},$$

where $c$ is a numerical constant, we used (15) for the second line, and a simple calculation to obtain the last line. Using Lemma 6, we obtain the desired conclusion.

# 6 Proof of Theorem M-4.1

In this section we will show that for $C$ large enough, if $\mathcal{A}$ holds, we have:

$$R(\hat{g}) - R^* \lesssim \epsilon. \tag{16}$$

Let $i^*$ be an arbitrary element of $\mathcal{S}^*$. Denote $T_i$ the number of queries required to eliminate an expert $i \in [\![K]\!]$. $T_i$ is upper bounded by the number of data points needed to have: $\Delta_{i^*i} > 0$, which would lead to the elimination of $i$ by $i^*$. The following claim, which is a consequence of Lemma 7, provides this upper bound.

**Claim 3.** *If $\mathcal{A}$ holds, let $i \in [\![K]\!]$ be a suboptimal expert ($\Lambda_i < +\infty$). We have:*

$$T_i \le 289 \log\left(KC\delta^{-1}\right)\Lambda_i.$$

*Proof.* Lemma 5 shows that experts $i^* \in \mathcal{S}^*$ are never eliminated if $\mathcal{A}$ is true. Using Lemma 7, the number of queries required for the elimination of a suboptimal expert $i$ by expert $i^*$, satisfies:

$$T_i \le 289 \log\left(KC\delta^{-1}\right)\Lambda_i.$$

$\square$

Let $\epsilon \ge 0$. Recall that $\mathcal{S}_\epsilon$ is defined by:

$$\mathcal{S}_\epsilon := \left\{ i \in [\![K]\!] : \Lambda_i > \frac{1}{\epsilon} \right\}$$

Suppose that we have:

$$C > 578 \left( \sum_{i \in \mathcal{S}_\epsilon^c} \Lambda_i + |\mathcal{S}_\epsilon| \min\left\{\frac{1}{\epsilon}; \Lambda^*\right\} \right) \log\left( K\delta^{-1}\left( \sum_{i \in \mathcal{S}_\epsilon^c} \Lambda_i + |\mathcal{S}_\epsilon| \min\left\{\frac{1}{\epsilon}; \Lambda^*\right\} \right) \right),$$

We therefore have using Lemma 2:

$$C > 289 \log\left(KC\delta^{-1}\right) \left( \sum_{i \in \mathcal{S}_\epsilon^c} \Lambda_i + |\mathcal{S}_\epsilon| \min\left\{\frac{1}{\epsilon}; \Lambda^*\right\} \right).$$

Let us denote by $C_1$ the total number of queries received by all the experts in $\mathcal{S}_\epsilon$ and by $C_2$ the total number of queries received by the remaining experts. We therefore have: $C = C_1 + C_2$. In order to show that at a certain round, all the experts in $\mathcal{S}_\epsilon^c$ were eliminated, it suffices to prove that:

$$C_1 \ge |\mathcal{S}_\epsilon| \max_{i \in \mathcal{S}_\epsilon^c} T_i,$$

since the inequality above shows that the budget is not totally consumed after round $\max_{i \in \mathcal{S}_\epsilon^c} T_i$ where all elements in $\mathcal{S}_\epsilon^c$ where eliminated.

Claim 3 provides the following upper bound for $C_2$:

$$C_2 \le 289 \log\left(KC\delta^{-1}\right) \sum_{i \in \mathcal{S}_\epsilon^c} \Lambda_i.$$

We therefore have:

$$C_1 = C - C_2$$

$$\ge 289 \log\left(KC\delta^{-1}\right) \left( \sum_{i \in \mathcal{S}_\epsilon^c} \Lambda_i + |\mathcal{S}_\epsilon| \min\left\{\frac{1}{\epsilon}; \Lambda^*\right\} \right) - C_2$$

$$\ge 289 \log\left(KC\delta^{-1}\right) \left( \sum_{i \in \mathcal{S}_\epsilon^c} \Lambda_i + |\mathcal{S}_\epsilon| \min\left\{\frac{1}{\epsilon}; \Lambda^*\right\} \right) - 289 \log\left(KC\delta^{-1}\right) \sum_{i \in \mathcal{S}_\epsilon^c} \Lambda_i.$$

Hence:

$$C_1 \geq 289 \log\left(KC\delta^{-1}\right)|\mathcal{S}_\epsilon| \min\left\{\frac{1}{\epsilon}; \Lambda^*\right\} \tag{17}$$

Recall that by definition of $\mathcal{S}_\epsilon$, using Claim 3 we have:

$$\max_{i \in \mathcal{S}_\epsilon^c} T_i \leq 289 \log\left(KC\delta^{-1}\right) \min\left\{\frac{1}{\epsilon}; \Lambda^*\right\},$$

hence:

$$C_1 \geq |\mathcal{S}_\epsilon| \max_{i \in \mathcal{S}_\epsilon^c} T_i.$$

This shows that $S \subseteq \mathcal{S}_\epsilon$. We have two possibilities: if $\frac{1}{\epsilon} < \Lambda^*$, the selected pair $(F_{\bar{k}}, F_{\bar{l}}) \in S \times S$ satisfies:

$$T_{\bar{k}\bar{l}} = \min\{T_{\bar{k}}, T_{\bar{l}}\} \geq \frac{C_1}{|\mathcal{S}_\epsilon|}.$$

Using (17), we have:

$$T_{\bar{k}\bar{l}} \geq 289 \log\left(KC\delta^{-1}\right)\frac{1}{\epsilon}. \tag{18}$$

Observe that Lemma 6 applies in this setting. In particular, the total number of rounds $T$ of algorithm M-1, satisfy: $T \leq C$. Hence, it holds

$$R\left(\frac{F_{\hat{k}} + F_{\hat{l}}}{2}\right) - R^* \leq c\,B\,\frac{\log(KC\delta^{-1})}{T_{\bar{k}\bar{l}}}.$$

We conclude by injecting inequality (18) in the bound above. We therefore have:

$$R(\hat{g}) - R^* \leq cB\,\epsilon,$$

where $c$ is an absolute constant.

If $\frac{1}{\epsilon} > \Lambda^*$, by definition of $\Lambda^*$ and the fact that $\mathcal{S} \subseteq \mathcal{S}_\epsilon$, we conclude that only the optimal experts (i.e. the experts $i$ such that $R_i = R^*$) remain when the budget is totally consumed. Hence combining any 2 of these expert will lead to the bound: $R(\hat{g}) \leq R^*$.

## 7   Proof of lower bounds

The lemma below gives a lower bound for the problem of estimating the parameter describing a Bernoulli random variable.

**Lemma 8** ([1], Lemma 5.1). Suppose that $\alpha$ is a random variable uniformly distributed on $\{\alpha_-, \alpha_+\}$, where $\alpha_- = 1/2 - \epsilon/2$ and $\alpha_+ = 1/2 + \epsilon/2$, with $0 < \epsilon < 1$. Suppose that $\xi_1, \ldots, \xi_m$ are i.i.d $\{0, 1\}$-valued random variables with $\mathbb{P}(\xi_i = 1) = \alpha$ for all $i$. Let $f$ be a function from $\{0, 1\} \rightarrow \{\alpha_-, \alpha_+\}$. Then it holds:

$$\mathbb{P}(f(\xi_1, \ldots, \xi_m) \neq \alpha) > \frac{1}{4}\left(1 - \sqrt{1 - \exp\left(\frac{-2\lceil m/2 \rceil \epsilon^2}{1 - \epsilon^2}\right)}\right).$$

### 7.1   Proof of Lemma M-6.1

Let $T > 0$ and consider an convex combination of experts $\hat{g}$ output after full observation of $T$ training rounds. We will construct two experts $F_1$ and $F_2$ and a target variable $Y$ and we will show that, for these variables, a strategy for our problem ($m = 2$ and $p = 1$) gives a solution to the problem in Lemma 8. Finally we will use the lower bound from this lemma.

For $\theta \in [0, 1]$, let $\mathbb{P}_\theta$ denote the probability distribution of $T$ i.i.d. draws $Y_1, \ldots, Y_T$ of Bernoulli variables or parameter $\theta$, while $F_{1,t} = 0$ and $F_{2,t} = 1$ almost surely for $t \in \llbracket T \rrbracket$. Let $\alpha$ be a variable that is uniformly distributed on $\{\alpha_-, \alpha_+\}$ with $\alpha_\pm = \frac{1}{2} \pm \frac{\epsilon}{2}$, and $\epsilon \in (0, 1)$ is a parameter to be tuned subsequently; let the training obervations be drawn according to $\mathbb{P}_\alpha$. Since $p = 1$, the output $\hat{g}$ is either $F_1$ or $F_2$. Define $f : \{0, 1\}^T \rightarrow \{\alpha_-, \alpha_+\}$ such that given $(Y_1, \ldots, Y_T)$, $f$ outputs $\frac{1}{2} - \frac{\epsilon}{2}$ if $\hat{g} = F_1$ and $\frac{1}{2} + \frac{\epsilon}{2}$ if $\hat{g} = F_2$. By construction we have that the events $\{f = \alpha\}$ and

$\{R(\hat{g}) = \min\{R_1, R_2\}\}$ are equivalent. Using Lemma 8 and setting $\epsilon = \frac{c_0}{\sqrt{T}}$ where $c_0$ is a constant such that the lower bound in Lemma 8 is equal to 0.1, we have:

$$\mathbb{P}\left(R(\hat{g}) - \min\{R_1, R_2\} \geq \frac{c_0}{\sqrt{T}}\right) > 0.1.$$

Due to the randomization of $\alpha$, the above probability is the average of the corresponding event under $\mathbb{P}_{\alpha_-}$ and $\mathbb{P}_{\alpha_+}$. Therefore, under at least one of these two training distributions, the deviation event has a probability at least 0.05.

## 7.2    Proof of Lemma M-6.2

The gist of the proof is the following. We will construct a distribution with two experts that are very correlated. In this situation, going from a weighted average of the two experts to a single expert with the largest weight does not change the prediction risk much, and so we could find a single expert with small risk if the weighted average has small risk. On the other hand, since the agent only observes one expert per training round, from their point of view the observational distribution is identical as if the experts were independent – the correlation cannot be observed. Therefore the same strategy could be used to find the best expert in the independent case. This contradicts the lower bounds in this case (which is a standard bandit setting), therefore it is impossible to pick consistently a weighted average with small risk in a situation where the correlations cannot be observed.

Let $T > 0$ be fixed. We consider the particular setting where the target variable $Y$ is identically 0, and the expert predictions $F_1$ and $F_2$ are two (non independent) Bernoulli random variables. We define a distribution $\mathbb{P}_-$ for $(F_1, F_2)$ such that:

- the marginal distribution of $F_1$ is Bernoulli of parameter $\alpha_- = \frac{1}{2} - \frac{\epsilon}{2}$;
- the marginal distribution of $F_2$ is Bernoulli of parameter $\alpha_+ = \frac{1}{2} + \frac{\epsilon}{2}$;
- it holds that $\mathbb{P}_-(F_1 F_2 = 1) = \alpha_-$.

Note that this can be easily constructed as $F_1 = \mathbf{1}\{U \leq \alpha_-\}; F_2 = \mathbf{1}\{U \leq \alpha_+\}$, where $U$ is a uniform variable on $[0, 1]$. Let $\mathbb{P}_+$ be defined similarly with the role of $F_1$ and $F_2$ reversed. Here, $\epsilon$ is a positive parameter to be tuned later. We denote $R_-, R_+$ for the prediction risks under distributions $\mathbb{P}_-, \mathbb{P}_+$. We have $R_-(F_1) = R_+(F_2) = \alpha_-, R_-(F_2) = R_+(F_1) = \alpha_+$, and $R^* = \alpha_-$ is the same under $\mathbb{P}_-$ and $\mathbb{P}_+$.

Let us be given an arbitrary training observation strategy $\pi$ (prescribing at each training round which expert to observe based only on past observations), and output a convex combination of experts $\hat{g}$. This output is a convex combination of $F_1$ and $F_2$, hence it is characterized by the weight of $F_1$, which we denote $\hat{\alpha}$. The parameter $\hat{\alpha}$ depends on the observed data. We also define $\hat{f}$ associated to this training strategy, that outputs $F_1$ if $\hat{\alpha} > \frac{1}{2}$ and $F_2$ otherwise. Finally, let us denote $\mathbb{Q}_\pi^+$ the distribution of the training data observed by the agent when the $T$ experts opinions are drawn i.i.d. from $\mathbb{P}_-$ and the agent observes the expert advices following strategy $\pi$; and define $\mathbb{Q}_\pi^-$ similarly.

Define the event $\mathcal{A}_+ := \left\{R_+(\hat{g}) - R^* \geq \frac{1}{4}\epsilon\right\}$ and similarly $\mathcal{A}_-$. In the remainder of the proof, we will show, using Bretagnolle-Hubert inequality (Theorem 14.2 in [2]), that either $\mathbb{Q}_\pi^-(\mathcal{A}_-)$ or $\mathbb{Q}_\pi^+(\mathcal{A}_+)$ is lower bounded by a positive constant.

We have under the distribution $\mathbb{P}_-$:

$$R_-(\hat{g}) - R_-(\hat{f}) = \mathbb{E}_-\left[(\hat{\alpha}F_1 + (1 - \hat{\alpha})F_2)^2\right] - \mathbb{E}_-\left[\left(\mathbb{1}\left(\hat{\alpha} > \frac{1}{2}\right)F_1 + \mathbb{1}\left(\hat{\alpha} \leq \frac{1}{2}\right)F_2\right)^2\right]$$

$$= \epsilon(1 - \hat{\alpha})^2 - \epsilon\left(1 - \mathbb{1}\left(\hat{\alpha} > \frac{1}{2}\right)\right)$$

$$\geq -\frac{3}{4}\epsilon.$$

Note that the above estimate crucially depends on the fact that $F_1, F_2$ are not independent under $\mathbb{P}_-$. In view of the above, the event $\mathcal{A}_-$ is implied by $R_-(\hat{f}) - R^* = \epsilon$. Similarly, $\mathcal{A}_+$ is implied by

$R_+(\hat{f}) - R^* = \epsilon$. Hence:

$$\mathbb{Q}_\pi^-(\mathcal{A}_-) + \mathbb{Q}_\pi^+(\mathcal{A}_+) \geq \mathbb{Q}_\pi^-\left(R_-(\hat{f}) - R^* = \epsilon\right) + \mathbb{Q}_\pi^+\left(R_+(\hat{f}) - R^* = \epsilon\right)$$
$$= \mathbb{Q}_\pi^-\left(\hat{f} = F_2\right) + \mathbb{Q}_\pi^+\left(\hat{f} \neq F_2\right).$$

Now we use Bretagnolle-Hubert inequality:

$$\mathbb{Q}_\pi^-(f = F_2) + \mathbb{Q}_\pi^+(f \neq F_2) \geq \frac{1}{2}\exp\left(-D\left(\mathbb{Q}_\pi^-, \mathbb{Q}_\pi^+\right)\right),$$

where $D(\mathbb{Q}_\pi^-, \mathbb{Q}_\pi^+)$ is the relative entropy between $\mathbb{Q}_\pi^-$ and $\mathbb{Q}_\pi^+$. In order to conclude, we need an upper bound on $D(\mathbb{Q}_\pi^-, \mathbb{Q}_\pi^+)$. Since the agent only observes one expert in each round according to strategy $\pi$, the distribution of the observed data $\mathbb{Q}_\pi^-$ or $\mathbb{Q}_\pi^+$ is unchanged if we replace the generating distributions $\mathbb{P}_-$ or $\mathbb{P}_+$ by distributions having the same marginals, but for which $F_1$ and $F_2$ are independent. Therefore, the observational distributions $\mathbb{Q}_\pi^-, \mathbb{Q}_\pi^+$ are equivalent to that of the observational distributions, under the same strategy, of a canonical bandit model with two arms. We can then use the divergence decomposition formula (Lemma 15.1 of [2]) to upper bound $D(\mathbb{Q}_\pi^-, \mathbb{Q}_\pi^+)$; denoting $\mathbb{P}_-^{(1)}, \mathbb{P}_-^{(2)}$ the marginals of $\mathbb{P}_-$ and similarly for $\mathbb{P}_+$, it holds

$$D\left(\mathbb{Q}_\pi^-, \mathbb{Q}_\pi^+\right) = \mathbb{E}_-[T_1]D(\mathbb{P}_-^{(1)}, \mathbb{P}_+^{(1)}) + \mathbb{E}_-[T_2]D(\mathbb{P}_-^{(2)}, \mathbb{P}_+^{(2)}),$$

where the expectation $\mathbb{E}_-[.]$ is with respect to the probability distribution $\mathbb{Q}_\pi^-$ and $T_i$ denotes the total number of rounds where the advice of expert $F_i$ was queried using the strategy $\pi$. We have: $T_1 + T_2 = T$ almost surely, and $D(\mathbb{P}_-^{(1)}, \mathbb{P}_+^{(1)}) = D(\mathbb{P}_-^{(2)}, \mathbb{P}_+^{(2)}) \leq 4\epsilon^2$ provided $\epsilon \leq \frac{1}{2}$. Therefore:

$$\mathbb{Q}_\pi^-(\mathcal{A}_-) + \mathbb{Q}_\pi^+(\mathcal{A}_+) \geq \frac{1}{2}\exp\left(-4\epsilon^2 T\right).$$

This shows that there exists a probability distribution $\mathbb{P} \in \{\mathbb{P}_-, \mathbb{P}_+\}$ for the experts advices and the target variable such that the prediction $\hat{g}$ satisfies:

$$\mathbb{P}(R(\hat{g}) - R^* \geq \epsilon) \geq \exp\left(-4\epsilon^2 T\right),$$

We conclude by choosing $\epsilon = \frac{1}{2\sqrt{T}}$.

# 8   Intermediate case: $m \geq 3, p = 2$

In this section we assume that the learner is allowed to access more than two experts advices per round. We show that this leads to an improvement of the bound in Theorem M-5.2. We consider the following extension of Algorithm M-2:

---
**Algorithm 1** Intermediate case
---
**Input** $m$, $L$ and $\rho$.
Initialization: $S \leftarrow [\![K]\!]$.
**for** $T = 1, 2, \ldots$ **do**
    Sample a subset $\mathcal{M}$ of size $m$ from $[\![K]\!]$ uniformly at random.
    Query the advice of experts in $\mathcal{M}$ and update the corresponding quantities.
    For all $i, j$: If $\Delta'_{ij} > 0$: $S \leftarrow S \setminus \{j\}$.
**end for**
**On interrupt:** Let $\hat{k} \in S$ and let $\hat{l} \leftarrow \underset{j \in S}{\operatorname{argmax}}\ \hat{d}_{\hat{k}j}$.
Return $\frac{1}{2}\left(F_{\hat{k}} + F_{\hat{l}}\right)$.

---

**Theorem 9.** (Instance independent bound) Suppose Assumption M-1 holds. Let $T \geq 1$, and denote $\hat{g}$ the output of Algorithm 1 with inputs $(m, L, \rho)$ in round $T$. If $m \geq 3$, then with probability at least $1 - \delta$:

$$R(\hat{g}) \leq \min_{i \in [\![K]\!]} R_i + cB\,\frac{(K/m)^2 \log\left(2TK\delta^{-1}\right)}{T},$$

where $c$ is an absolute constant.

*Proof.* Let $i, j \in [\![K]\!]$, denote $T_{ij}(T)$ the total number of rounds where the advice of expert $i$ and $j$ were jointly queried. We have: $T_{ij}(T) = \sum_{t=1}^{T} \mathbb{1}\{i \text{ and } j \text{ were jointly queried at round } t\}$. We conclude that $T_{ij}(T)$ is the sum of $T$ independent and identically distributed Bernoulli variables with parameter: $\frac{m(m-1)}{K(K-1)}$. We therefore have the following consequence of Bernstein concentration inequality, with probability at least $1 - \delta$, for all $i, j \in [\![K]\!]$ and $T \geq K$:

$$|T_{ij}(T) - \mathbb{E}[T_{ij}(T)]| \leq \sqrt{2T\frac{m(m-1)}{K(K-1)}\log(2KT/\delta)} + \frac{1}{3}\log(2KT/\delta). \qquad (19)$$

Suppose that $\delta$ satisfies:

$$\log(2KT/\delta) \leq \frac{1}{16}\frac{m^2}{K^2}T.$$

Then we have:

$$\sqrt{2T\frac{m(m-1)}{K(K-1)}\log(2KT/\delta)} + \frac{1}{3}\log(2KT/\delta) \leq \frac{1}{2}\frac{m(m-1)}{K(K-1)}T, \qquad (20)$$

Observe that the result of Lemma 6 still holds in this setting for non-eliminated elements (experts in $S_T$), since the elimination criterion for an expert $j$, which consists of the existence of $i$ such that $\Delta'_{ij} > 0$, is the same as in Algorithm M-2. Let $\hat{g}$ denote the output of Algorithm 1, we conclude that if $\mathcal{A}$ and (19) hold for all $i, j$ and $T$, we have:

$$R(\hat{g}) - R_{i^*} \leq \kappa \frac{\log(KT\delta^{-1})}{T_{\hat{k}\hat{l}}(T)}, \qquad (21)$$

where $\kappa$ is a constant depending only $\eta, L$ and $\rho$. Finally, we use (20). We therefore have with probability at least $1 - 4\delta$:

$$R(\hat{g}) \leq \min_{i \in [\![K]\!]} R_i + c\,B\frac{(K/m)^2\log(2TK\delta^{-1})}{T}.$$

Now suppose that $\delta$ satisfies:

$$\log(2KT/\delta) \geq \frac{1}{16}\frac{m^2}{K^2}T,$$

then it holds:

$$\frac{(K/m)^2\log(2TK\delta^{-1})}{T} \geq \frac{1}{16}.$$

We conclude that for $\bar{c} = \max\{c, 16\}$ we have:

$$R(\hat{g}) - \min_{i \in [\![K]\!]} R_i \leq B \leq \bar{c}B\frac{(K/m)^2\log(2TK\delta^{-1})}{T}.$$

$\square$