# OpenReview forum: "Fast rates for prediction with limited expert advice"
_NeurIPS.cc/2021/Conference — NeurIPS 2021 Poster_

### Official Review · Reviewer_4EKs · 2021-07-06

**Rating:** 8
**Confidence:** 4

**Summary:**

UPDATE: The authors have addressed my questions in their response, and confirmed my initial positive view of the paper. Their plan for improving the presentation of the introduction seems satisfactory.

---

The paper considers aggregation of a finite set of K predictors
(experts) in the (batch) statistical learning setting, when the loss is
Lipschitz and strongly convex. It presents learning algorithms that
perform one sequential pass through the training data, and satisfy one
of the following constraints:
* no constraint: full information
* budget constraint: algorithms can query at most C experts in total
  during their run
* query and output constraint: algorithms can query at most m experts
  per round, and output a convex combination of at most p experts. The
  following specific cases are considered:
  - m=p=2 (and extension m>2,p=2)
  - m=1 and p=2
  - m=2 and p=1.
In all cases (near-)optimal bounds are obtained in probability.


**Ethical Concerns:**

No ethical issues.

**Limitations And Societal Impact:**

Yes. This is a theoretical paper, for which limitations have been adequately discussed.
Societal impact is too indirect to discuss.

**Main Review:**

Strengths:

The result for the full information case alone justifies acceptance. This
setting is really of fundamental importance, because of its application
to model selection (see [3] and [17]) and because it is known that
neither ERM over the predictors nor ERM over the convex hull of the
predictors achieves the optimal rate (see [17]). Thm 1 in the paper does
achieve the optimal rate previously obtained by [3] and [17], but with a
much simpler and easier to understand analysis. At first sight, this
does appear to come at the cost of the assumption that the loss is
strongly convex, which is much stronger than the assumptions in [3] and
does not cover many cases of practical interest, but inspection of the
proof appears to show that this assumption can be relaxed to assuming
only strong convexity of the risk, which is a much more liberal
assumption. -- Can the authors comment on whether this is correct?

The other types of constraints considered in the paper are all
variations on the approach for the full information case, but each of
them requires an additional new idea and corresponding non-trivial
choice of algorithm.

All in all, there are multiple new ideas in this paper that will be of
interest to the learning community.

Weaknesses:

Unfortunately, the presentation of Sections 1 and 2 could be improved:

Most notably, I found section 1 to be very vague, both with respect to
motivation and related work (note that it contains no references at
all!) and in the sense that it did not contain sufficient technical
details of the setting, which is not quite standard. I therefore could
not understand the discussion of related work in section 2 before
reading the formal definition of the setting in section 3.

The reference to 'expert advice' in the terminology, while defensible,
is perhaps not ideal, since this is not the standard prediction with
experts setting. Terminology and motivation in [3] and [17] seem much
clearer to me. I also get the impression that you are secretly thinking
of experts as having chosen a function f_i from some function class and
making predictions by F_{i,t} = f_i(X_t) based on some underlying
feature vector X_t. This is the only way I can link the discussion in
Section 7 to [18] and make sense of the statement that the "closure of
the experts class is non-convex".

Minor comments:

There are occasional grammatical mistakes. For instance, in the statement of Theorem 1. While the meaning is clear, it would be good to do another round of proofreading before the final version.

In (2): supremum should be w.r.t. i instead of j.

Proof of Thm 1, second displayed equation: why is there a division by 8
in the first inequality? I would expect a division by 2 based on strong
convexity. It would also be convenient to define strong convexity via
the inequality that you actually use here.

Proof of Thm 1: please spell out the last step of maximizing over
d_{\bar k, \bar j} in sufficient detail that the reader can verify the
result without redoing the calculations themselves.

End of Section 5, remarks: this discussion is incorrect, because for
epsilon=0 the set S_epsilon^c does contain i such that Lambda_i =
infinite. (I suspect that this will be fixable.)

Section 6: it shouldn't be necessary for the notation \hat{R}_{ij}(j,t)
to refer to 'j' twice.

Algorithm 3: based on the discussion, I would expect the argmin in the
selection of j to be restricted to the set S. (Note that the same argmin
also appears in the discussion below Algorithm 3.)

Corollary 4: Please mention/discuss worse dependence on K compared to
Theorem 1.

Theorem 5: Please state explicitly that the result holds for any fixed
epsilon.

Section 7: when citing [18], it is recommended to also cite the
correction in W. S. Lee, P. L. Bartlett and R. C. Williamson,
"Correction to “The importance of convexity in learning with squared
loss”", 2008. And possibly also S. Mendelson, "Lower bounds for the
empirical minimization algorithm", IEEE Transactions on Information
Theory, 2008.


**Time Spent Reviewing:**

5

---

> ### Author Response · Authors · 2021-08-09
> **Reply to 4EKs**
>
>
>
> We thank the reviewer for the positive and detailed review as well as the suggestions for improvement.
>
> To respond to some specific points:
>
> * *Presentation of Sections 1 and 2*
>
> We propose to shorten Section 1 and to swap the ordering of Sections 2 and 3 so that the discussion of previous work comes after the presentation of the formal setting.
>
> * *Concerning the "experts" terminology*
>
> The reviewer is right that we were initially thinking of the experts as predictors from an underlying feature vector X, and we will make this instantiation of the setting more prominent in the Setting section. (Admittedly in the current version the reference to model selection in Section 1 is too terse.)
>
> We decided to adopt the experts terminology because we felt that it would make readers coming from the community of (stochastic) bandits/experts more comfortable and facilitate the comparison to other works on limited feedback, found typically in that thread of literature. While [18] considered predictors, the distance they consider is the $L^2(P)$ distance; thus it makes sense to also speak of "closure of the set of experts" if those are random variables on the same underlying probability space, in the sense of $L^2$ distance. (Of course this point of [18] is only relevant for an infinite class of experts/predictors anyway, which is not the setting we consider here.)
>
> * *About the assumption of strong convexity:*
>
> Thank you (and the other reviewers) for raising this point. It is correct that we only need the strong convexity of the risk function with respect to the $L^2$ distance (we still need the $l(.,y)$ to be L-Lipschitz a.s. for all $y$); however, under this weaker condition we need to add loss boundedness as an assumption, in order to get an empirical Bernstein's inequality which constitutes the main probabilistic tool for our results (weaker moment assumptions ensuring the validity of an empirical Bernstein's inequality would also do).
>
> More generally, we think that it is possible to adapt our analysis to the assumptions made in [3] or in [17], the objective being to enable the risk function to enjoy an inequality similar to the parallelogram equality for the quadratic function.
>
> * *Proof of Thm 1, second displayed equation: why is there a division by 8 in the first inequality?*
>
> A more detailed development for this inequality is presented on page 5 of the supplementary material (in the line after (7)).
>
> * *End of Section 5, remarks: this discussion is incorrect, because for $\epsilon=0$ the set $S_\epsilon^c$ does contain $i$ such that $\Lambda_i = \infty$*. (I suspect that this will be fixable.)*
>
> Thank you for spotting that inconsistency. The statement of Theorem 2 must be amended to distinguish the case when $\mathcal{S}_\epsilon$ only contains optimal experts, in which case the term $|\mathcal{S}_\epsilon|/\epsilon$ can be dropped from the bound. This applies also to Theorem 5 (instance dependent bound for two queries per round). (For simpliciy, we have not written this explicitly in the new version of Theorem 5 above.)

---

### Official Review · Reviewer_pnAv · 2021-07-16

**Rating:** 7
**Confidence:** 3

**Summary:**

In this paper, the authors consider the problem of prediction with limited expert advice, with the goal of minimizing the excess generalization error, in the stochastic regime. They study a setting where they assume that the loss function is L-Lipschitz and \rho strongly convex.

They show that if the learner is only allowed to either make one observation per round in the prediction phase or query the advice of only one expert in the test phase, then there is a probability bounded by a constant that the worst case excess risk has a slow convergence.
To circumvent this issue, they study the impact of changing the number of observations and the number of expert queries, showing that in particular using at least two observations and two expert queries instead of one is sufficient to achieve a fast rate convergence of the excess generalization error.
They consider a budgeted setting, where the learner has global budget constraint, and a more classic limited expert's predictions setting, which fixes the number of observed expert advices at each round.




**Limitations And Societal Impact:**

Concerning the limitations, I could not find a proper discussion on whether Assumption 1 is reasonable in practice.
The interest of using limited advice was properly discussed, and efficient ressource management can be seen as a positive social impact.

Besides that, the societal impact was not really discussed, but it does not necessartily make sense for such a theoretical problem.

**Main Review:**

This paper is overall quite complete, as the authors study two versions of the prediction with limited expert advice problem, one with a global budget, and one with a per round fixed budget. They do also provide lower bounds which justify the interest of using two queries per round rather than 1.
Furthermore, both problem dependent and problem independent bounds are provided.
A special case of the budgeted setting also relates to the problem of best arm identification in multi-armed bandits, where the assumptions on the loss function and the capacity to query several arms at each round are used to achieve improvements compared to the standard Best arm identification results that are subjected to a stricter problem setting.

Overall, the problem seeting seems novel and the contributions significant as they prove some fast rate convergences.  This being said, I don't know whether the assumption that the loss function is L-Lipschitz and \rho strongly convex is realistic.

==== After Rebuttal changes
After reading the rebuttal, I rose the rating to a 7.

**Time Spent Reviewing:**

6

---

> ### Author Response · Authors · 2021-08-09
> **Reply to pnAv**
>
> Thank you for your valuable feedback.
> See the reply to reviewer 4EKs for a discussion on some specifc points.

---

### Official Review · Reviewer_bhrD · 2021-07-16

**Rating:** 6
**Confidence:** 3

**Summary:**

The setting in the paper is a prediction with expert advice in the stochastic setting, while the learner observes a limited number of experts each round (denoted by m), and is limited on how many experts to choose in prediction (denoted by p).
The main result is providing an algorithm with fast rates convergence if 1/T, as long as the learner is allowed to see two experts per round and to use two experts for prediction.
Moreover, for m=p=1 the authors provide a lower bound of 1/sqrt{T}.


**Limitations And Societal Impact:**

-

**Main Review:**

Significance: The setting of expert advice is classic and important.
Extending it with limited access seems also interesting and natural.
The results show a separation -- what is the amount of access necessary for the learner in order to achieve fast generalization rates.

Originality/novelty:
The main algorithm/technique seems novel to me.
However, I'm not familiar with results on other related models with limited feedback.

Clarity and relation to prior work:
The literature review is thorough.
Altogether, the overall method is presented clearly.

**Time Spent Reviewing:**

3

---

> ### Author Response · Authors · 2021-08-09
> **Reply to bhrD**
>
> Thank you for reviewing our paper.
> See the reply to reviewer 4EKs for a discussion on some specific points.

---

### Official Review · Reviewer_YBCx · 2021-07-17

**Rating:** 7
**Confidence:** 2

**Summary:**

The paper proposes a novel learning framework, which seems to me to be a cross between prediction with expert advice and bandit framework ported to stochastic off-line settings.

The key features are:

1. On every round, the learner chooses a limited number of experts to observe.
2. After T rounds, the learner outputs a function on experts' predictions that to be used on the next round.
3. The experts and the outcome are stochastic.


The framework seems to naturally fit a scenario where obtaining an expert prediction (e.g., by running a computationally expensive program) is costly.

The upper and lower bounds on the expectation of the learner's loss on the (T+1)th round are provided (lower bounds without proofs). The cases where m, the number of experts we are allowed to observe, is equal to the total and less than the total are treated separately.

**Ethical Concerns:**

No.

**Limitations And Societal Impact:**

Yes.

**Main Review:**

I think this is an interesting learning paradigm and the paper provides a thorough investigation.

**Time Spent Reviewing:**

2

---

> ### Author Response · Authors · 2021-08-09
> **Reply to YBCx**
>
> Thank you for your review and positive feedback.
> Please note that the proof of the lower bounds are presented in section 7 of the supplementary material.

---

### Author Response · Authors · 2021-08-09
**A modification in theorem 5 [Instance dependent bound for two queries per round]**

---
title: ""
output: html_document
---
We thank the reviewers for their appreciation and their remarks.

In this common rebuttal part we must report that we have recently identified an unfortunate mistake in the proof of Theorem M-5 (supplementary material, proof of claim 3, page 9: the pivot $j$ might be eliminated early by a non-optimal expert, so that the lower bound on the time $T_{ji^*}$ is not valid).

Fixing this issue requires modifying Algorithm M-3; we present here an alternative along with a new instance dependent bound (very similar to the previous one) to replace theorem M-5. All other results still hold. We hope that given the limited scope of this change this can be acceptable.

In the modified Algorithm M-3 we do not use the pivot strategy any longer, but simply sample equally all couples of experts in the current pool $S$.

**Modified Algorithm 3: Two-point feedback**

| **INPUT:** $\delta, L$ and $\rho$.
| **INITIALIZATION:** $S \gets [K]$.
| **FOR** T=1,2,... :
|          Let $(i,j) \in arg\min_{(u,v) \in S \times S} T_{uv}$.
|          Query the advice of experts $i$ and $j$ and update the corresponding quantities.
|          For all $u,v$: If $\Delta_{uv}'>0$: $S \gets S \setminus \{v\}$.
| **END FOR**;
| **On interrupt:**
| Let $k \in S$, $l \in arg \max \left \lbrace d_{kj}; j \in S\right \rbrace$
| Let $q$ denote the empirical risk minimizer on $S$
| **IF:** $T_{kl} > \sqrt{\log(KT\delta^{-1}) T_q}$
|           **Return** $\frac{1}{2} \left(F_{k} + F_{l}\right)$.
| **ELSE:**
|     **Return** $F_{q}$.

**Theorem M-3** and **Corollary M-5** (empirical and instance independent bounds) remain true. Namely, the event $\mathcal{A}$ holds with probability $1-\delta$ for any strategy. On this event, (a) the optimal expert $i^*$ remains in $S$ at all times; and (b) Lemma S-6 therefore still holds (at any time) for any experts $i,j$ that remain in the pool $S$, since all couple of experts remaining in $S$ have been sampled the same number of times (up to a difference of 1). The proof of Theorem M-3 and Corollary M-5 is then unchanged.

**Theorem 5 [Instance dependent bound for modified Algorithm 3 ]**
	Suppose Assumption 1 holds and let $\epsilon >0$. Let $\hat{g}$ denote the output of Algorithm 3 with input $(\delta, L, \rho)$ and $T$ denote the total number of rounds. If :
$$
T \ge 578 C_\epsilon \log (\delta^{-1} C_\epsilon),
$$
where
$$
C_\epsilon :=  K \sum_{i \in S_\epsilon^c}\Lambda_{i} + 2 \frac{\left|\mathcal{S}_{\epsilon}\right|^2}{\epsilon},
$$
then with probability at least $1-\delta$ it holds $R(\hat{g})  \le R^* + cB \epsilon,$ where $c$ is an absolute constant.

**Proof:**
We use Lemma S-6 which, as we argued above, is still valid with the new strategy. It is therefore sufficient to prove that for any $i,j \in S_T$, it holds $T_{ij}(T) \geq B \log(K \delta_T^{-1})/\epsilon$ under the assumption made on $T$; this will give the conclusion.

Let $N_{\epsilon}$ denote the integer satisfying: $\Lambda_{(N_{\epsilon})} \leq \frac{1}{\epsilon} < \Lambda_{(N_{\epsilon}+1)}$, so that $|S_\epsilon| = K - N_{\epsilon}.$ Define the following sequences $$\phi_N := 289(K-N)^2(\Lambda_{(N)}-\Lambda_{(N-1)} )\log(\delta^{-1}C_\epsilon);$$
$$\qquad \tau_N := \sum_{k=1}^N \phi_k.$$
	Recall that the number of queries of required to eliminate an expert $i\in [K]$ is upper bounded by the number of samples for the couple $(i,i^*)$ needed to have: $\Delta_{ii^*}>0$, which would lead to the elimination of $i$ by $i^*$. The minimum number of samples for this at time $t$ is upper bounded via the first claim of Lemma S-7 (which remains valid regardless of the strategy), namely $289 \log(K\delta_t^{-1}) \Lambda_{i}$. From this, it can be proved easily by induction that on the event $\mathcal{A}$, for any $N\leq N_\epsilon$, after round $\lceil\tau_N\rceil$, (a) all experts $i$ satisfying $\Lambda_{i} \le \Lambda_{(N)}$ are necessarily eliminated, so that there are at most $(K-N)$ experts remaining; and  (b) all couples of experts remaining in $S$ have been sampled at least $289 \log(\delta^{-1} C_\epsilon) \Lambda_{(N)}$ times.
	Furthermore, standard algebra gives for any $N \in [K]$:
		$$
		\tau_{N} = 289 \log\left(\delta^{-1} C_\epsilon \right) \left(   \sum_{i=1}^{N-1}(2(K-i) +1) \Lambda_{(i)} + (K-N)^2\Lambda_{(N)}\right).
		$$
	Under the assumed condition on $T$, it can be seen that $T-\lceil\tau_{N_\epsilon} \rceil \geq 289 \log(\delta^{-1} C_\epsilon) (K-{N_\epsilon})^2 \epsilon^{-1}$.
	Since after time $\lceil\tau_{N_\epsilon}\rceil$ there was at most $(K-{N_\epsilon})$ experts in $S$, we conclude that: $T_{\hat{k}\hat{l}}(T) \ge 289\log(KT\delta^{-1}) /\epsilon$, and apply the first claim of Lemma S-6 to obtain the result.

---

### Decision · Program_Chairs · 2021-09-27

**Decision:**

Accept (Poster)

**Comment:**

The reviewers are excited about the results of the paper and unanimously recommend acceptance.